# A 2D hybrid perovskite ferroelectric with switchable polarization and photoelectric robustness down to monolayer

Yuzhong Hu [1,2] ✉, Haidong Lu [3], Shehr Bano Masood[3], Clemens Göhler [2], Shangpu Liu[4], Alexei Gruverman [3] & Marin Alexe [1] ✉

The continuous dimensional scaling of semiconductor and logic photoelectric device requires ferroelectrics to possess robust photoelectric activity and switchable polarization at the nanoscale. However, traditional ferroelectrics such as oxide perovskites generally suffer from relatively large bandgap and deteriorated ferroelectricity in ultrathin forms, while the polarization in many transition metal dichalcogenides is related to inter-layer effects, leading to ferroelectricity that only exists in flakes with a certain layer number and particular stacking forms. The associated challenging fabrication and high-cost synthesis of inorganic ferroelectrics currently render mass industrial production of ultrathin ferroelectric semiconductors impossible. Here with $(isopentylammonium)_2(ethylammonium)_2Pb_3I_{10}$, we report an organic-inorganic hybrid perovskite nanoflake with cheap solution synthesis, switchable polarization, a narrow bandgap (1.86 eV to 2.21 eV form bulk to monolayer), and robust photoelectric properties down to the monolayer. The present work reveals the great potential of 2D hybrid perovskite ferroelectrics as low-cost ferroelectric semiconductors at the nanoscale.

Historically, ferroelectrics were considered dielectrics with switchable polarization due to their generally wide bandgap and consequent electrically insulating nature. The early works on ferroelectric semiconductors in the 1960s, especially SbSI showing robust photoresponse, led to a tide of discoveries of important semiconductor properties of ferroelectrics such as bulk photovoltaic effect[1,2] and significantly broadened their applications[3,4]. The innate hysteretic properties of the polarization give these materials the capacity to transform single-state electronics and photoelectronics into logic/memory devices. For example, semiconductor properties such as conductivity and photoconductivity of ferroelectric semiconductors can be manipulated by the number of domain walls[5], while optical excitations such as polarized light can be used to switch ferroelectric polarization[6].

However, as semiconductor and optical devices continue to scale down, the integration of ferroelectrics in ultrathin forms becomes crucial. Yet, materials with switchable polarization and resilient photoelectric activity at thickness of few naometers are still rarely to be seen. This generally results in either deteriorated ferroelectricity or a relatively weak photoelectric signal in ferroelectric nanodevices[7,8]. Perovskite oxide thin films are the most intensively studied ferroelectric system. However, they generally have a wide band gap, usually above 3 eV, and relatively poor semiconductor properties[9]. Additionally, the increasing depolarization field with thickness scaling and their 3D nature lead to detrimental polarization[10] and the formation of a materials/substrate interface layer with defective ferroelectricity at the nanoscale[11]. 2D inorganic structures such as some transition metal dichalcogenides (TMDs) flake systems have recently been discovered

[1]Department of Physics, The University of Warwick, Coventry, UK. [2]Institute for Molecular Systems Engineering and Advanced Materials, Heidelberg University, Heidelberg, Germany. [3]Department of Physics and Astronomy, University of Nebraska Lincoln, Lincoln, NE, USA. [4]Physikalisch-Chemisches Institut, Universität Heidelberg, Heidelberg, Germany. ✉e-mail: huyu0012@e.ntu.edu.sg; m.alexe@warwick.ac.uk

to be ferroelectric. However, the ferroelectricity in many of these systems is related to specific 2D stacking forms or interlayer effects[12]. Ferroelectricity will disappear with either for different layer stacking (bilayer h-BN, WSe$_2$ and MoSe$_2$, etc.) or for different layer number/thickness (WTe$_2$, $\beta'$-In$_2$Se$_3$ and layer-dependent ferroelectricity in $\alpha$-In$_2$Se$_3$, etc.)[11,12]. For the device fabrication, it is thus necessary to first conduct thickness measurements/identification and then selectively pick up films with the target thickness from all exfoliated flakes. Additionally, the costly facilities required for chemical vapor deposition and pulsed laser deposition of inorganic ferroelectrics, along with the expenses associated with high-temperature and energy supply, significantly increase production costs. These factors make the synthesis and fabrication of 2D inorganic ferroelectrics particularly challenging and expensive, rendering mass industrial production of ultrathin ferroelectric semiconductors currently impossible[11,13].

Van der Waals (VdW) hybrid perovskite ferroelectrics (HPFs) is an emerging member of the ferroelectric family with a chemical formula of A$_2$B$_{n-1}$C$_n$X$_{3n+1}$. Here the small organic cation (B) and metal-halide octahedron (CX$_6^{2-}$) build the 3D network, which is encapsulated between 2D layers consisting of the large organic spacer "A". These spacer layers are bonded to each other by VdW forces and thus induce 2D features[14]. This distinguishes them from HPFs with layered structures that are bonded by stronger chemical interactions, such as hydrogen and covalent bonds in structures like (BDA)(EA)$_2$Pb$_3$Br$_{10}$[15] and (EATMP)PbBr$_4$[16]. Similar to other hybrid perovskite materials, single crystals and thin films of HPFs can be obtained through low-cost and convenient techniques such as solution crystal synthesis and spin-coating[17]. These facile processing methods can significantly reduce production costs and potentially make the commercialization of semiconductor devices feasible in the future[18]. However, most previous studies on VdW HPFs have focused on discovering new compositions and investigating bulk crystals[19], such as research on the bulk semiconductor properties of (BA)$_2$PbCl$_4$[20] and piezoelectricity of (R/S-BrBA)$_2$PbBr$_4$[21]. Although more than seventy 2D organic-inorganic hybrid ferroelectrics have been reported, their nanoscale properties are still almost hiding in the dark. We propose they shall be excellent ferroelectric semiconductors at the nanoscale. Specifically, the intrinsic polarization originating from off-center ordering of molecular cations and anions in each single-layer building block determinates there is no issue of stacking form or layer number dependent ferroelectricity as seen in many TMDs systems. Meanwhile, the VdW bounded layer structure and in-plane polarization shall mitigate issues related to substrate/ferroelectric interface straining and increasing depolarization field at the nanoscale. With optically active 3D parts such as PbI$_6^{2-}$ as building block, which are the origins of the active photovoltaic properties in hybrid perovskite solar cell materials[14,22,23], the flakes of VdW HPFs may also possess intrinsic photoelectric robustness.

(isopentylammonium)$_2$(ethylammonium)$_2$Pb$_3$I$_{10}$ (PEPI) is a typical and newly discovered VdW HPF, characterized by a single-layer unit of IPA-EAPbI-IPA bonded by VdW force[24]. Here we reveal the great potential of VdW HPFs as a promising material family as nano-ferroelectric semiconductors and report the robust ferroelectricity as well as photoelectric activity in exfoliated PEPI films down to the monolayer limit. Remarkably, the switchable ferroelectricity survives at the nanoscale, while the photoconductivity ($4.03 \times 10^{-3}$ and $9.81 \times 10^{-5}$ S m$^{-1}$ for 75 nm and monolayer flakes) are several orders higher than those of oxide ferroelectric semiconductors (generally $10^{-9}$ to $10^{-6}$ S m$^{-1}$, see Table 1). The intrinsic (layer-number-independent) ferroelectricity distinguishes HPFs from many 2D inorganic ferroelectrics. This indicates that crystal flakes and spin-coated thin films are in principle ferroelectric regardless of their specific thickness, suggesting that the complex device fabrication processes required for many VdW inorganic materials are unnecessary for HPFs. The demonstrated combination of robust photoelectric activity, facile processing, and intrinsic ferroelectricity is rarely observed within the ferroelectric family. This positions PEPI and VdW HPFs as ideal candidates for low-cost and practical 2D ferroelectric semiconductors, and they may serve as important building blocks for functional heterostructures. The employed nanoscale Hetch equation can be used to estimate photoelectric robustness (by calculating the product of carrier mobility $\mu$ times lifetime $\tau$) for other in-plane nanoflake systems.

## Result and discussion
### Synthesis and basic characterizations
Millimeter-size PEPI single crystals (see Supplementary Fig. 3a) were obtained by slow evaporation (see Methods). Characterizations were first carried out to confirm the material's composition, exfoliable feature and bulk crystal ferroelectricity. The composition was characterized by powder X-ray diffraction (XRD) (Supplementary Fig. 3b) and differential scanning calorimetry (Supplementary Fig. 3c, and see Supplementary Fig. 4 for the corresponding phase transition of this biaxial ferroelectric), both of which show good agreement with the previous report[24]. The exfoliable structure with a single layer building block of IPA-EAPbI-IPA (EA and IPA are ethylammonium and isopentylammonium, respectively) with a thickness of 2.81 nm can be identified from the crystallographic structure[24] of PEPI (see Fig. 1a). The exfoliable feature was confirmed by atomic force microscopy (AFM) imaging of flakes exfoliated on glass slide by Scotch tape (see Fig. 1b–d for optical microscope image, AFM topography image and the corresponding height profile, respectively). The surface topography shows step structures similar to typical 2D materials. The thickness of the monolayer is identified as 3.05 nm, which is in good agreement with the reported crystallographic result[24]. The ferroelectricity of this material was measured on bulk crystal with crystal orientation identified by pole figure XRD measurement (Supplementary Fig. 3d). Along the polar direction (see Fig. 1e for an example of two states along the $c$ axis), a remnant polarization of around 5.5 μC cm$^{-2}$ was obtained at room temperature with a coercivity of about 12 kV cm$^{-1}$ under 100 Hz bipolar electric field (Fig. 1f), which is consistent with the previous report[24].

### Piezoelectric force microscope characterizations
As ferroelectricity is a collective parameter involved in interlayer interactions in bulk, a key question for ferroelectric thin flake is whether the spontaneous polarization can survive effects such as thermal fluctuation and substrate clamping when its dimension reaches the molecular thickness limit. We exfoliate PEPI thin flakes onto Au-coated silicon substrate and conducted piezoelectric force microscope (PFM) studies to characterize their ferroelectric polarization. The working principle of PFM in-plane polarization mapping is illustrated in Fig. 2a–c. Under AC voltage, a radial electric field is applied on the flake sample (Fig. 2a). The shear component of the piezoelectric effect induces torsional motion in the tip, with magnitude indicated by the PFM amplitude signal, and polarization direction illustrated by the phase signal. The ferroelectric domain structure can thus be mapped by scanning across the flake (Fig. 2b) while polarization reversal can be realized by line sweeping of the tip with a DC voltage applied (Fig. 2c).

Based on this method, PEPI flakes with various thicknesses were scanned for mapping. As shown in Fig. 2d–i, clear multi-domain feature can be observed in flakes with different thicknesses. By comparing topography and phase diagrams, it is obvious that the height and phase signal show weak correspondence, indicating the origin of the phase contrast is a difference in polarization direction rather than topology. As the flake thickness decreasing, the amplitude becomes smaller while the phase barrier becomes slightly noisier (Fig. 2i and Supplementary Figs. 5–7). This thickness scaling induced change shall be attributed to the decreasing piezoelectric response as observed in many other 2D and conventional ferroelectric thin films[25–27], which is shall be attributed to nonuniform electric field from the AFM tip and

**Table 1 | Synthesis and photoelectric properties of different ferroelectric semiconductors**

| Materials | Synthesis | $T$ (nm) | $\sigma_P$ (S m$^{-1}$) | $J_P$ (A cm$^{-2}$) | $E_g$ (eV) | Reference |
|---|---|---|---|---|---|---|
| BTO | PLD | 20 | $1.75 \times 10^{-9}$ | $6.3 \times 10^{-6}$ | 3.31 | 53 |
| BZT-BCT | PLD | 700 | $2.99 \times 10^{-7}$ | $4.7 \times 10^{-5}$ | 3.2 | 54 |
| PLZTN | MOD | $4 \times 10^5$ | $4.8 \times 10^{-9}$ | $3.7 \times 10^{-8}$ | 2.5 | 55 |
| BFO | PLD | 150 | $4.97 \times 10^{-6}$ | $4.97 \times 10^{-4}$ | 2.70 | 56 |
| BBLT | SSM | $5 \times 10^5$ | $2.03 \times 10^{-9}$ | $6.5 \times 10^{-9}$ | 3.2 | 57 |
| SbSI | HDT | $10^5$ | $9.54 \times 10^{-5}$ | $3.18 \times 10^{-5}$ | 1.83 | 33 |
| PEPI | Solution | 75 | $4.03 \times 10^{-3}$ | $2.43 \times 10^{-1}$ | 1.98 | This work |
| PEPI | Solution | 3 | $9.81 \times 10^{-5}$ | $3.65 \times 10^{-3}$ | 2.21 | This work |

PLD, MOD, HDT and SSM refer to pulsed laser deposition, metalorganic deposition hydrothermal and solid-state method, respectively. Parameters $T$, $\sigma_P$, $J_P$, and $E_g$ refer to thickness, photo-conductivity, photocurrent density and bandgap, respectively.

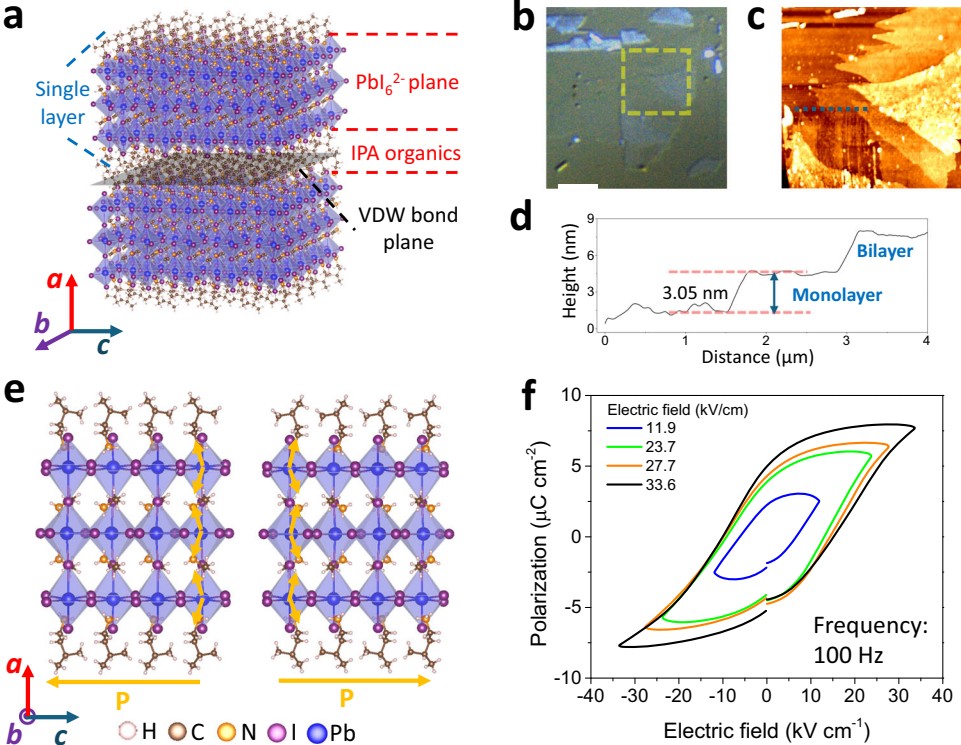

**Fig. 1 | Structure and ferroelectric property of $(IPA)_2(EA)_2Pb_3I_{10}$.**
**a** Crystallographic structure. The EA$^+$ cation and PbI$_6^{2-}$ octahedral compose the 3D part while the IPA$^+$ is the large organic spacer as the 2D part. Optical microscope image (**b**, the scale bar is 4 μm), AFM topography image (**c**) and the corresponding height profile (**d**) of few-layer $(IPA)_2(EA)_2Pb_3I_{10}$ along the dotted line. **e** Ferroelectric states with opposite directions (left and right) along the $c$-axis of a single layer

$(IPA)_2(EA)_2Pb_3I_{10}$. With the N$^+$ in the organic parts and the center of PbI$_6^{2-}$ regarded as cation and anion centers, electric dipoles can be identified as orange arrows. The vertical projections offset each other, and in-plane polarization can be identified along the $c$ directions. **f** Electric field-dependent ferroelectric hysteresis loops along the polar direction of the bulk crystal.

possible increasing clamping effect from the surrounding[28,29]. However, multidomain feature with clear phase contrast can be identified in very thin flakes (Fig. 2f, i), which demonstrates the survival of ferroelectric polarization. To confirm the piezoelectric origin of the PFM signal, we conducted a temperature-dependent PFM study. Supplementary Fig. 8 shows the PFM results of a PEPI flake at room temperature and at 343 K (PEPI transitions to the paraelectric phase at 340 K). The flake exhibits an obvious multidomain structure and decent PFM amplitude at room temperature. However, both features disappear at 343 K, i.e., above the phase transition. The strong correlation between the PFM signal and the phase transition provides solid evidence that the PFM signal primarily originates from piezoelectric activity rather than other effects, such as electrostatic interaction.

Deteriorated ferroelectricity and even pinned polarization is a crucial issue in ultrathin ferroelectric systems[30]. To investigate the polarization reversal property and further confirm the ferroelectricity in PEPI nanoflakes, PFM switching investigations were carried out on these flakes. As indicated in Fig. 2c, due to the radial electric field, polarization on two sides of the sweeping track shall be switched to opposite directions. As shown in Fig. 2d–f (see topology and amplitude images in Supplementary Figs. 5–7), biases are applied sequentially with the tracks indicated by the red lines. The consistent sequential polarization switching confirms the switchable ferroelectricity of PEPI down to monolayer. We noticed that sometimes the sweeping leaves a thin dent along its track (see Supplementary Fig. 6, topography diagram of the first switch). This shall be due to the mechanical soft nature

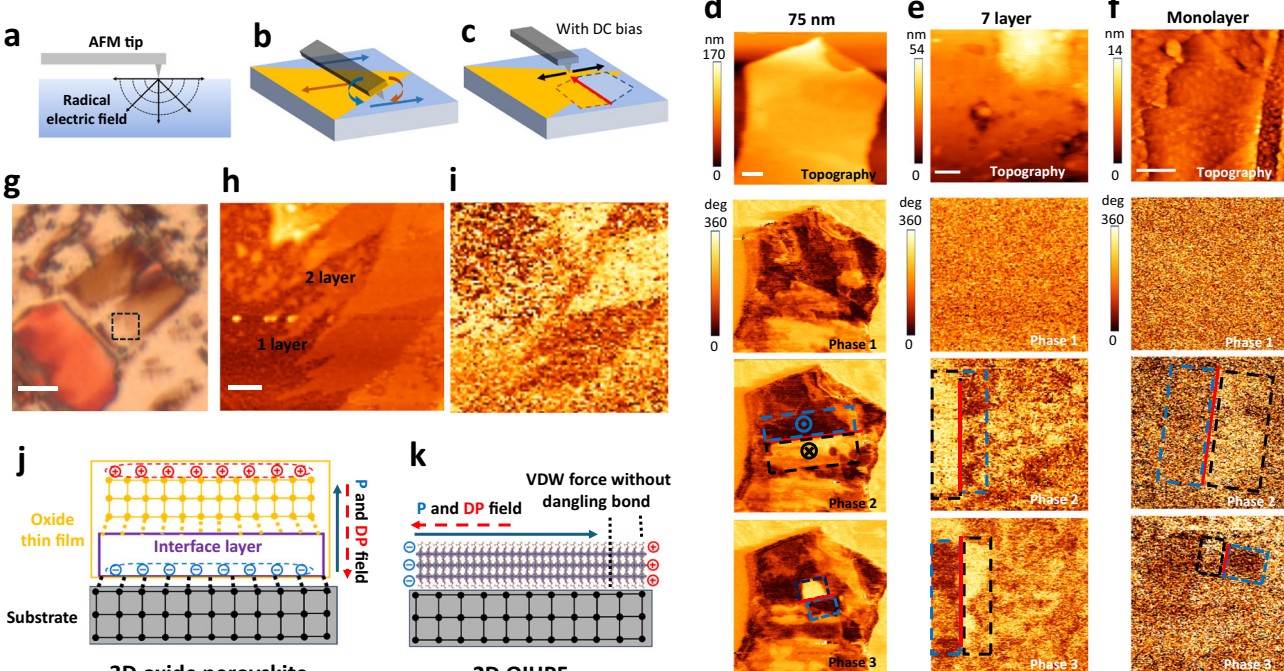

**Fig. 2 | PFM characterization of the ferroelectric domain structures and polarization switching on PEPI thin flakes. a** Radial electric field in thin flake with bias from the tip. **b** Electric field induced torsional movement on the cantilever due to ferroelectric polarization while scanning. The blue/brown curved arrows indicate the torsion in different directions due to corresponding polarization in blue/brown arrow directions. **c** Polarization reversal induced by PFM tip sweeping (red arrow) with DC bias. The black arrows indicate the electric field perpendicular to the sweeping direction. **d**–**f** Topography and polarization switching in $(IPA)_2(EA)_2Pb_3I_{10}$ flakes with different thicknesses. The scale bars are 1, 0.5 and 0.5 μm for the topography images of (**d**–**f**) respectively. The color scales for all phase diagrams are unified. The tip-sweeping tracks and switched areas with polarization in opposite directions are indicated by the red line, blue and black dash boxes, respectively. Photo (**g**, black dash box indicates the AFM imaging area, scale bar is 8 μm), topography (**h**, scale bar is 0.5 μm) and phase images (**i**) of few-layer flakes. The structures and depolarization fields in oxide thin film (**j**) and VdW HPFs flake (**k**). Orange and black dotted line: mechanical strains due to lattice mismatch in corresponding interfaces. Blue and red dashed arrows: polarization and depolarization field. Red and blue circles: Negative and positive charges due to screening effect.

of the 2D hybrid structure, which generally has a Young's modulus one to two orders smaller than those of oxide ferroelectrics[31]. The survival of ferroelectric polarization in the ultrathin flakes of this VdW HPFs shall be attributed to its VdW structure and in-plane polarization nature. As indicated in Fig. 2j, k, for conventional oxide ferroelectric thin films with out-of-plane polarization, the deterioration of ferroelectric polarization is mainly attributed to the depolarization field along the thickness direction and the pining effect at the substrate-ferroelectrics interface due to intrinsic lattice mismatch. These issues, however, are not the case for VdW HPFs. In these hybrid ferroelectric structures, the films show a dangling-bond-free tail, and the connection with the substrate is via VdW forces, of which bonding energy is generally two orders weaker than that of the covalent bonds at oxide heterostructure interfaces[32]. This effectively screens mechanical strain from the interface and isolates the dipoles in 2D hybrid ferroelectrics from the surroundings. At the same time, the in-plane oriented polarization of VdW HPFs generates a much weaker depolarization field perpendicular to scaling down direction, which shall be in principle independent of thickness change.

## Photoelectric characterizations of bulk crystals

The photoelectric properties of PEPI were then investigated first on bulk crystals. Au electrodes (0.25 mm electrode distance) were deposited on the sample to study wavelength and temperature dependence of photoconductivity (see Fig. 3a). A narrow effective band gap of 1.86 eV comparable to SbSI (1.83 eV)[33] can be identified from the spectrum at room temperature (see Supplementary Fig. 1b and supplementary note 1 for details). As the temperature decreases to

77 K, the band gap increases almost monotonously to 1.97 eV, while the photocurrent in the short-wavelength region shows an obvious increase (Supplementary Fig. 1d). This increase shall be attributed to a decrease in surface recombination[34], which may be induced by the widened band gap and increasing tilting in the inorganic framework at low temperature, leading to a longer excited-state lifetime and suppressed recombination coming from Pb−I vibrations[35]. Power-dependent photocurrent measurement was then carried out on a crystal with the same device structure along polarization direction under 520 nm laser illumination. As shown in Fig. 3b, the obtained dark current and photocurrent density ($J$) at 15 V bias are 0.24 and 610 μA cm$^{-2}$, respectively. With uniform illumination, a large photocurrent density of 8.38 mA cm$^{-2}$ and a high on/off ratio of around $3.5 \times 10^4$ were obtained. The dark and photocurrent along the out-of-plane direction are nevertheless several orders smaller (see Fig. 3c, 0.012 and 0.13 μA cm$^{-2}$, respectively). The large anisotropies in both dark and illuminated conditions indicate the 2D large organic layer (IPA$^+$) is much more electrically insulating than the inorganic layer, while the robust photoelectric property along the in-plane direction shall be attributed to the 2D lead-iodide plane, which is generally characterized by long carrier life and high carrier mobility[36]. The produce of carrier mobility ($\mu$) and lifetime ($\tau$) of a photoelectric system with illumination direction normal to the electric field can be estimated by fitting $J$–$V$ (current–voltage) characteristics to the Hetch equation[37,38]:

$$J = \frac{J_0 \mu \tau V}{w^2}\left[1 - \exp\left(\frac{-w^2}{\mu \tau V}\right)\right] \quad (1)$$

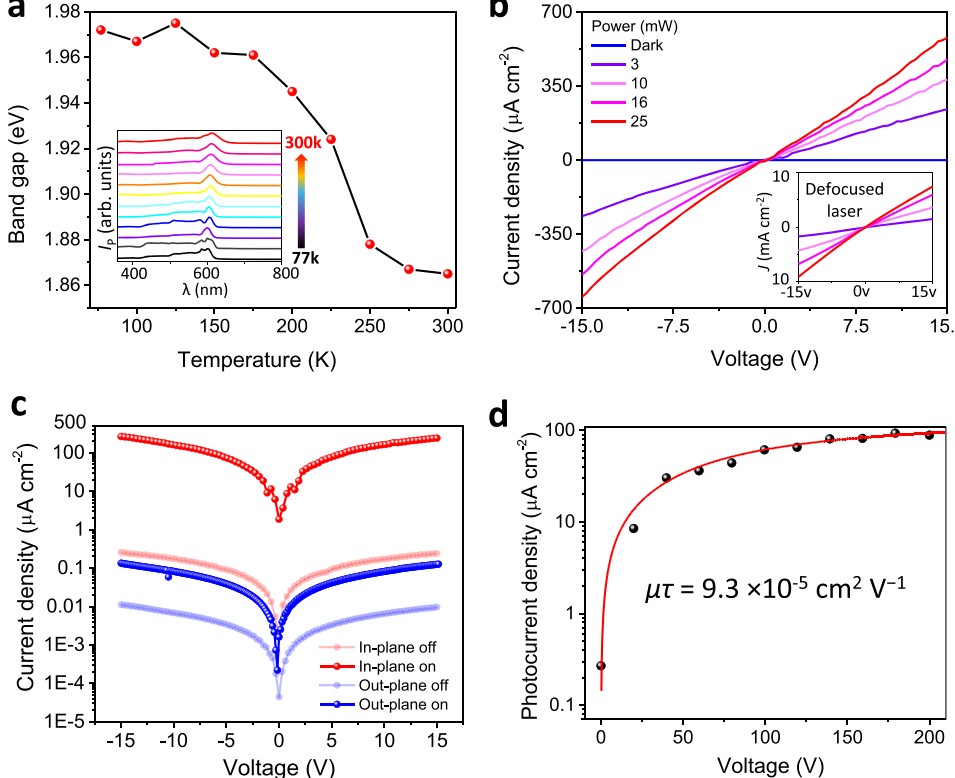

**Fig. 3 | Photoelectric properties of bulk crystal. a** Temperature-dependent band gap extracted from photocurrent ($I_P$) spectra (inset). **b** Photocurrent density measured by 520 nm laser with various powers. The inset shows the $J–V$ curves under uniform illumination (defocused laser). Curves obtained under the same laser power in (**b**) and this inset are labeled with the same color. The photocurrent shows a significant increase. This is primarily due to the larger illumination area and photoexcited carriers near the electrodes (where focused illumination cannot cover) are with shorter diffusion lengths. **c** Comparison between the photocurrent and dark current density along the in- and out-of-plane directions under a 520 nm laser with power of 7 mW. **d** $J–V$ characteristics used for the $\mu\tau$ product estimation. The red line is the fit using Eq. 1.

where $J$, $J_0$, $\mu$, $\tau$, $V$ and $w$ are photocurrent density, saturated photocurrent density, carrier mobility, carrier lifetime, applied voltage and device width (inter-electrode distance), respectively. This equation governs the photoresponse-electric bias behavior of materials and is widely used in various semiconductor systems for the estimation of photoelectric robustness[37,39,40]. Materials with larger $\mu\tau$ values show signs of saturation at a smaller electric field, indicating the photo-excited carriers can move fast and survive long enough to reach the electrode even under a small potential difference between electrodes. The $\mu\tau$ product of $9.3 \times 10^{-5}$ cm$^2$ V$^{-1}$ obtained for this VdW HPFs is similar to those of other layered hybrid perovskites[41,42], demonstrating the intrinsic robust photoelectric activity of this ferroelectric.

## Photoelectric characterizations of thin flake

The photoelectric properties at the nanoscale were then investigated on crystal flakes with different thicknesses. PEPI thin films were first exfoliated from bulk crystal by Scotch tape and transferred by polydimethylsiloxane (PDMS) to a glass slide with prepatterned Au electrodes enabling applying the electric field parallel with polarization direction (see Fig. 4a for photos of 75 nm flake and monolayer). The device structure of this in-plane photoelectric system is shown in Fig. 4b. Thickness-dependent photoconductivity spectrum was first characterized to investigate potential size effects. As indicated in Fig. 4c, with dimension decreasing from above micrometer to monolayer, the energy band gap extracted from the spectrum increases monotonously from 1.86 to 2.21 eV. Similar band gap increases with dimension scaling down were also observed in other lead-iodide hybrid perovskites[43] and TMDs[44], which shall be attributed to the increase in exciton binding energy and change in the edge state of conduction band[44,45]. $J–V$ characteristics were first measured on 75 nm flake under 520 nm illumination with various power. As shown in Fig. 4d, under 8 and 10 mW laser, the photocurrent increases first linearly with voltage and shows signs of saturation at voltage around 3.5 V. This saturation at low bias indicates the large $\mu\tau$ of the PEPI flake. The dark current and photocurrent densities at 5 V are 5.35 $\mu$A cm$^{-2}$ and 0.243 A cm$^{-2}$, respectively, producing a high on/off ratio of $4.5 \times 10^4$ and giant photoconductivity of $4.03 \times 10^{-3}$ S m$^{-1}$.

We next investigated the thickness-dependent photoelectric activity of PEPI flake. As shown in Fig. 4e, the photoresponse was measured under the same illumination power with thickness scaling down to monolayer, revealing an obvious decrease in photoelectric robustness. In the monolayer flake, the obtained photocurrent density and photoconductivity at 4 kV cm$^{-1}$ are 3.65 mA cm$^{-2}$ and $9.81 \times 10^{-5}$ S m$^{-1}$, respectively (see Fig. 4e and Supplementary Fig. 9), which however, are still orders of magnitude higher than those of classic ferroelectrics semiconductors including BFO and BTO (Fig. 4f). The superior photoelectric activity of PEPI as a nanoscale ferroelectric semiconductor can be indicated in Table 1.

## The Hecht equation for nanodevices

Finally, we discuss the decline in photoresponse in ultrathin PEPI flakes and the Hetch equation for nanoscale devices. Besides the blue shift of the photoconductivity spectrum (Fig. 4c) and a corresponding decrease in light absorption, another important factor in photocurrent decline shall be the decrease in $\mu\tau$ especially in carrier lifetime. This is expected as when the dimension of the flake approaches the molecular limit, the light-excited electrons and holes will be confined in almost a 2D space with a thickness of only few nanometers. This dimensional

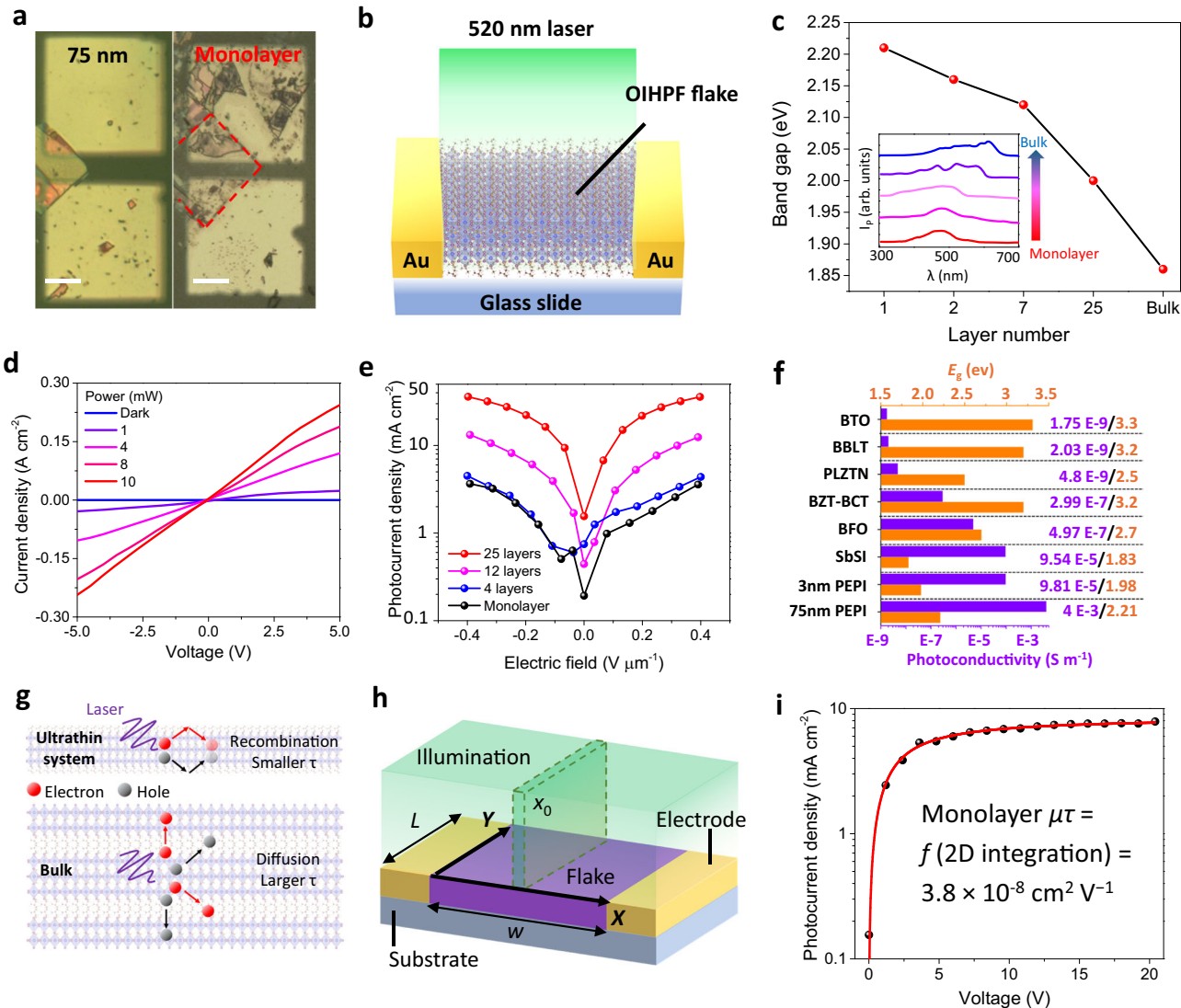

**Fig. 4 | Photoelectric properties of crystal flakes. a** Photos of typical 75 nm and monolayer devices with scale bars representing 12 and 15 μm, respectively. **b** Schematic of the flake device structure under illumination. **c** Thickness-dependent band gap extracted from photocurrent ($I_P$) spectra (inset). **d** J−V characteristics of 75 nm PEPI flake under a 520 nm laser with various power. **e** Layer-dependent J−E characteristics measured at a constant power of 2 mW. **f** Photoconductivity and bandgap ($E_g$) comparisons of different ferroelectric materials. **g** Schematic of electron/hole recombination in bulk and ultrathin systems. **h** Physics parameters, coordinates and illumination for $\mu\tau$ calculation of the 2D flake system. The illumination part outlined by the green dashed line with a darker color indicates an example of an illumination strip (a differential of the whole illumination area) and its position along x direction ($x_0$). The w and L are the sample width and electrode length, respectively. **i** J−V characteristics of monolayer flake for $\mu\tau$ estimation.

confinement will lead to a dramatic increase in recombination due to carriers' frequent reflection at the material boundary (see Fig. 4g). Additionally, under this condition, carriers will also become more sensitive to surface defects due to more frequent reflection and contact[46]. This decrease in $\mu\tau$ can be directly reflected by the more linear J−V curves of the monolayer device (Supplementary Fig. 9), which show no sign of saturation like those of the 75 nm flake.

However, for J−V curves obtained in monolayer PEPI and other flake materials, we noticed that a quantitative estimation of $\mu\tau$ using the classic Hetch Eq. (1) is not appropriate. Specifically, one of the boundary conditions of (1) is that the illumination area should be much smaller than the device. Consequently, in the derivation of the class equation, the light-excited carriers that reach the electrodes can be regarded as originating from a point/strip of generation with identical drifting length equal to the illumination point/strip-electrode distance[38,39]. However, the dimensions of 2D flake-based nanodevices

in most cases are around or below 10 μm. This is much smaller than a generic laser spot in labs (at least few hundred micrometers for diode-produced lasers without an additional focusing lens), where the illumination area will cover the whole flake and generate carriers throughout the device. In this case, carriers that reach the electrode will have position-dependent rather than identical drifting length. This will largely change the development and form of the Hetch equation.

Here we analyze and discuss a nanodevice version of the Hetch equation, which can provide a rough estimation of $\mu\tau$ for flake devices with similar structure. The development process and employed principles are overall similar to the classic case[38] but use a different integration method due to the change in illumination conditions (see Supplementary note 2 for details). Briefly, the entire illumination area covering the device can first be divided into narrow strips with differential width of $dx_0$ as shown in Fig. 4h. For calculation simplicity, the shape of the device is assumed to be a cuboid with a thin thickness.

Under this condition, the local induced charge ($Q_0$,) at position $x_0$ is:

$$Q_0 = \eta N e \frac{L}{wL} dx_0 = \frac{\eta N e}{w} dx_0 \quad (2)$$

Where $\eta$, $N$, $w$, $L$ and $e$ are the quantum efficiency, number of photons reaching the flake area, width of the flake, length of the electrode and elementary charge, respectively. According to Lenard's relationship[38], under an electric bias $V$ and due to effective recombination and limited lifetime, the induced carriers by this strip will distribute across the device with their number decreasing exponentially along the drifting direction. The number of carriers at position $x$ can be described as:

$$Q_x = Q_0 \exp\left(-\frac{(x_0-x)w}{\mu\tau V}\right) = \frac{\eta N e}{w} dx_0 \exp\left(-\frac{(x_0-x)w}{\mu\tau V}\right) \quad (3)$$

By integrating the charging effect of all these carriers, and assuming the number of photons reaching this strip per unit of time to be $dN/dt = \Phi$, the current density induced by this strip can be obtained as (see supplement note 2 for detail):

$$J_{x0} = \frac{1}{TL}\int \frac{dQ_x}{dt}\frac{dx}{w} = \frac{\mu\tau V\eta\Phi e}{w^3 TL} dx_0\left(1-\exp\left(\frac{x_0 w}{\mu\tau V}\right)\right) \quad (4)$$

Here $T$ is the thickness of the flake. This modified Hetch equation has similarity in form to the classic one, as both calculate the photo-response at the electrode induced by a thin illumination strip but handle the illumination differential differently. The photocurrent density of the device can thus be obtained by integrating the strip throughout the entire illumination (device) area as:

$$J = \int d(J_{x0}) = \frac{\mu\tau V\eta\Phi e}{w^2 TL}\left(1+\frac{\mu\tau V}{w}\exp\left(-\frac{w^2}{\mu\tau V}\right)-\frac{\mu\tau V}{w^2}\right) \quad (5)$$

Equation (5) provides the relationship between $\mu\tau$ and $J$ of the nanodevice and shows very good fitting to our monolayer $J$–$V$ curve (see Fig. 4i). The resultant $\mu\tau$ of $3.8\times10^{-8}$ cm$^2$ V$^{-1}$ is much smaller than that of a bulk crystal. As discussed earlier, this dramatic decrease in $\mu\tau$ shall be attributed to the increase in recombination due to space confinement. Similar dimensional scaling effect on $\mu\tau$ with orders of decrease in magnitude have also been reported in other 2D systems such as $MoS_2$[46] and $WSe_2$[47].

To further demonstrate the accuracy and the applicability of the generalized Hecht equation across different materials system, we measured the photocurrent on polycrystal $MAPbBr_3$ and PM6:Y6 thin films with the same device structure and fitted the curve with Eq. (5) (see Supplementary Fig. 10). The obtained $\mu\tau$ value of $2.7\times10^{-7}$ cm$^2$ V$^{-1}$ on $MAPbBr_3$ film is highly consistent with those of previous reports[48,49]. We conducted the same characterization on PM6:Y6 thin films, a well-known organic solar cell material[50]. The obtained $\mu\tau$ value of $8.81\times10^{-9}$ cm$^2$ V$^{-1}$ also shows strong consistency with previous reports of ~$10^{-9}$ cm$^2$ V$^{-1}$ [51,52]. This data demonstrates the developed equation can be employed across different material systems.

In summary, we have addressed here two critical issues related to: (i) the scarcity of 2D ferroelectric materials, and (ii) the high costs associated with the synthesis and fabrication of ultrathin ferroelectric semiconductors. With PEPI, we demonstrate that VdW HPFs possess a combination of switchable ferroelectricity, facile processing, and robust photoelectric activity down to the monolayer limit, which is rarely observed among the ferroelectric family. The insights and demonstrations presented in this study significantly enrich the small family of 2D ferroelectrics and advance the development of practical ferroelectric semiconductor nanodevices. Furthermore, the employed generalized nanoscale Hetch equation extends the boundary condition limitations of the classic Hetch equation to nanoscale devices. It

provides an effective method for estimating $\mu\tau$ and and consequently quantify the robustness of photoconductivity in the case of nanoflake systems.

## Methods

### Synthesis of (isopentylammonium)$_2$(ethylammonium)$_2$Pb$_3$I$_{10}$ single crystal

Single crystals of PEPI were synthesized by the slow evaporation of a filtered solution consisting of hydroiodide acid (57 wt.%), iso-penylamine, ethylamine solution (2 M in methanol) and lead (II) acetate trihydrate (PbAC) at 296 K. Specifically, 2.3 g PbAC was first dissolved in 15.34 ml hydroiodide acid at 296 K, then 2574 µl of EA and 265 µl of IPA were slowly added to the solution in sequence. The solution was then stirred with a magnetic bar for 3 h, filtered, and left in an oil bath at 296 K for slow evaporation. Crystals in flake shape with millimeter-scale lateral dimensions and thicknesses of a few tens of micrometers were obtained after approximately three weeks of slow evaporation.

### X-ray diffraction characterization

Powder X-ray diffraction and pole figure measurements were conducted using Cu Kα radiation (λ = 1.540598 Å) on a commercial diffractometer (Panalytical Xpert PRO).

### Differential scanning calorimetry measurement

Differential scanning calorimetry characterization was carried out by a TA INSTRUMENTS - Q10 setup. The bulk crystal sample was heated and cooled with rate of 5 K min$^{-1}$ in a sealed aluminum pan under nitrogen flow at ambient pressure.

### Device fabrication

For flake devices used in photoelectric characterizations, Ti/Au (3/15 nm) electrodes were first deposited on glass slide by e-beam evaporation using a standard photolithography process and a shadow mask. Flakes were exfoliated from bulk single crystal with Scotch tape by attaching the tape to the $bc$ crystal plane and exfoliated along the out-of-plane direction. The flakes were then transferred on substrate by Polydimethylsiloxane (SYLGARD™ 184 Silicone Elastomer Kit, with ratio of 10:1) in a glove box to ensure a clean flake-electrode interface. For bulk crystal devices for in-plane photoelectric characterizations, 50 nm parallel Au electrodes with millimeter-level length and 0.25 mm inter-distance were deposited on single crystal using a shallow mask and e-beam evaporation. For out-of-plane characterizations, Au electrodes with millimeter-level side length were deposited on top and bottom sides of the crystal by the same method with square-shaped shadow mask. The thickness of the Au is 3 nm for good transparency.

### Photoelectric characterizations

The temperature-dependent photoconductivity spectra were measured using a cryostat (Janis VPF-700) and illuminating the samples with monochromatic light produced by a 100 W xenon lamp and a monochromator (Oriel, Cornerstone 130). The wavelength-dependent photocurrent was measured by an electrometer (Keithley, 6517B). Room temperature photocurrent under green laser (520 nm laser diode, THORLABS, L520P50) was measured by a source meter (Keithley, 2636B) on a probe station in vacuum. The electric field was applied parallel to the polarization direction by contacting the patterned Au electrodes.

### Atomic/piezoelectric force microscope characterizations

AFM and PFM characterizations were conducted by a XE-100 Park and Asylum Research AFM system equipped with a home-built current amplifier/filter system on flakes exfoliated on 50 nm Au-glass slides and silicon substrates. Au-coated silicon tips with a spring constant of ~0.2 N m$^{-1}$ (BudgetSensors ContGB-G) were used to avoid sample

surface damage. Flake samples are oriented with polarization perpendicular to the cantilever to enhance the piezoelectric signal. PFM images were obtained with a typical driven voltage of 1 V by both resonant (Asylum) and non-resonant modes (XE-Park). The tip sweeping speed and applied voltage for polarization reversal were typically 1–5 μm s$^{-1}$ and 4–10 V, respectively. The piezoelectric signal on some flakes can be weaker than others, which shall be attributed to the lateral clamping effect induced by the substrate.

### Ferroelectric characterization

Ferroelectric (PE loop) measurements were performed on a probe station by an aixACCT Ferroelectric tester at room temperature with a measurement frequency of 100 Hz.

## Data availability

The processed data in this study are available at Dryad with link: https://doi.org/10.5061/dryad.m905qfvc8.

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

## Acknowledgements

The financial support by the Engineering and Physical Science Research Council (EPSRC) EP/T027207/1 and EP/P025803/1 is gratefully acknowledged. We thank Michael Crosbie for technical support. We thank Brian Hinz for offering PM6:Y6 thin films for photocurrent measurements. We thank Dr. Han Shiguo and Dr. Liu Xitao for their valuable discussions on single crystal synthesis. Y.H. acknowledges the Alexander von Humboldt Foundation of Germany for the financial support for his research activities in Germany.

## Author contributions

Y.H. and M.A. conceived the idea and designed the project. H.L., S.B.M., and A.G. assisted with and conducted part of the PFM measurements. C.G. designed and conducted part of PM6:Y6 photocurrent measurements. S.L. assisted part of PEPI crystal synthesis. Y.H. conducted the rest of the experiments. H.L. contributed to part of the writing. Y.H. and M.A. wrote the manuscript with input from all authors.

## Competing interests

The authors declare no competing interests.
