## [Transparent Peer Review file · Nature Communications]

A 2D hybrid perovskite ferroelectric with switchable polarization and superior photoelectric robustness down to monolayer

Corresponding Author: Dr Yuzhong Hu

Version 0:

Reviewer comments:

Reviewer #1

(Remarks to the Author)

In this manuscript, the authors reported an organic-inorganic hybrid perovskite nanoflake with switchable polarization, a narrow bandgap and robust photoelectric properties down to the monolayer limit. Building on previous work, they further explored the impact of the size of ferroelectric thin films on the photoelectric effect and investigated the underlying mechanisms. Their research provides certain insights into the photoelectric applications of ferroelectric thin films. I suggest that the following minor modifications should be made before it can be published.

1 In the title "A van der Waals hybrid perovskite ferroelectric semiconductor", what is the "van der Waals hybrid perovskite". Here, in the Introduction, it should be introduced, and the introduction should provide the characteristics of this structure and similar structures reported. In this way, we can understand why you mentioned this word in such a prominent title.

2 In Fig. 1d, what does each line represent? Please label them.

3 In the DSC results (Fig. S3c), why are there two small peaks at 120°C and 50°C during the heating process? And it's best to use Kelvin temperature for all. Different types of samples were made to compare with Ref.21, Is there any impact or difference? Because the microscopic crystalline states are actually the same.

4 In Fig. S3b, why is the line in the PXRD result from experimental testing not flat? It is recommended to show in normalization.

5 You mention "PEPI thin films were first exfoliated from bulk crystal by Scotch tape", which crystal plane did you use to obtain the film? The carrier mobility may differ for different crystal planes.

6 finally, you mentioned that polarization switching in (IPA)₂(EA)₂Pb₃I₁₀ flakes with different thicknesses. First, I want to know the mechanism behind the difference in thickness, is it for conduction? Or a weak interaction? Or something else. Second, what is the impact of electric field action.

Reviewer #2

(Remarks to the Author)

Referee's report on "A van der Waals hybrid perovskite ferroelectric semiconductor with 1 switchable polarization and superior photoelectric robustness down to 2 monolayer" by Yunzhong Hu et.al.

The authors reported device testing on a perovskite ferroelectric semiconductor, named (isopentylammonium)₂(ethylammonium)₂Pb₃I₁₀ (PEPI). They observed switchable polarization and exceptional photoelectric durability reaching down to a thickness of just two monolayers. This organic-inorganic hybrid perovskite nanoflake shows switchable polarization, a bandgap ranging from 1.86 eV to 2.21 eV (from bulk to monolayer), and photoelectric characteristics even at the monolayer scale. However none of these data can be considered as unique or new in the light of what had been published in the field.

Consequently, I do not support the publication of this work in Nature Communications as the findings were very routine and there was no new physics or chemistry.

This work is “opportunistic” as it attempts to exploit the interest in 2D ferroelectrics but failed to show more than trivial data. The novelty of the work may be questioned concerning its distinctiveness compared to existing published literature in the field for several reasons:

First there is rather limited novelty in material design. The use of organic-inorganic hybrid perovskite materials and the exploration of their ferroelectric and photoelectric properties have been subjects of numerous research. The overall concept of using such hybrid materials in ferroelectric semiconductors has been previously explored.

The narrow bandgap and robust photoelectric properties down to the monolayer level is not so significant, similar properties have been reported in other 2D materials or material systems, albeit with variations in specific values. I don't see why this is considered novel, in view of the fact that depolarization effects in hybrid perovskites is less compared to 2D inorganic system because the former is more insulating and thicker!

The comparison with traditional oxide ferroelectric semiconductors in terms of photoconductivity is selective and biased. Photoconductivity is a function of absorption and also conductivity of the material and it is meaningless comparing to wide band gap system.

While the use of the nanodevice Hetch equation to estimate mobility-lifetime product and photoelectric robustness is a novel aspect of the work, the overall significance of this methodology in the broader context of material characterization and analysis needs to be further established.

The extent to which these findings significantly advance the field compared to existing research on similar materials or device architectures is highly questionable.

Reviewer #3

(Remarks to the Author)

Through the exfoliation of PEPI crystal, a van der Waal hybrid perovskite ferroelectric semiconductor, the authors successfully obtained thin nanoflakes that survive the ferroelectricity even down to the monolayer limit. Moreover, the authors demonstrated that the PEPI monolayer exhibits superior and robust photoelectric properties, with photoconductivity several orders higher than that of traditional oxide ferroelectric semiconductors. The manuscript demonstrates the promising potential of PEPI nanoflakes in the realm of nano-photoelectronic devices, illuminating new avenues for future optoelectronic applications. This study, therefore, presents some novelty in the field. After addressing the following comments, this work could be accepted for publication.

1.The inclusion of pioneering literature on Van der Waal hybrid organic-inorganic ferroelectric semiconductors is essential for a comprehensive understanding (Nat Commun 6, 7338 (2015)).

2.For the verification of the ferroelectricity in PEPI nanoflakes, especially for the monolayer sample, the PFM signal is too weak to judge a definite polarization reversal. The in-plane PFM signal correlates with the component of in-plane polarization projected onto the cantilever normal. Thus, it is suggested to reorient the sample orientation with respect to the PFM cantilever to enhance signal strength. In addition, given such a weak PFM response, the authors are advised to distinguish between contributions from ferroelectric switching and electrostatic influences.

3.It is suggested to incorporate corresponding color scales to Fig. 2b and c, as well as Supplementary Fig. 5 to 7, to enhance the interpretability of the data presented.

4.What are the measurement conditions for the different curves in Fig. 1d and the inset of Fig 3b.

5.The polarization of PEPI is along the in-plane direction. However, there exists an offset along the out-of-plane direction between dark and photocurrent curves, as illustrated in Fig. 3c. It means that the device has a current response in the out-of-plane off state at zero bias. I wonder what causes this offset and whether it is associated with the polarization direction.

Version 1:

Reviewer comments:

Reviewer #1

(Remarks to the Author)

It was well revised and can be accepted for publication.

Reviewer #2

(Remarks to the Author)

The authors have outlined their reasons for believing that demonstrating the ferroelectricity of single-layer perovskites is significant. However, I argue that this is not particularly interesting, as it is unsurprising that ferroelectricity can persist in thin

layers of such dielectric confined systems. For instance, even in purely inorganic materials like In_2Se_3 , ferroelectricity has been shown to survive down to the monolayer level. Ferroelectricity down to 1 nm has been demonstrated in Bismuth oxide. Science, (2023) Vol 379, Issue 663. Why should it be so special that a hybrid organic inorganic system that is easy to exfoliate will show single unit cell ferroelectricity? how is this useful ? This is not a film but a flake.

Additionally, the claim that hybrid perovskites differ based on synthesis costs is not particularly relevant and comes off as a trivial excuse. Simply reducing the material to thinner layers without any improvement in switching metrics compared to bulk—merely stating that polarization and coercive fields are similar to previous reports—demonstrates a lack of understanding. Furthermore, the compatibility of in-plane ferroelectricity with the two-terminal vertical devices required for ferroelectrics is not addressed, suggesting that this work is out of touch with the current field and may not benefit the community.

I do not find any new insights in this study, and the authors' responses have not provided substantial justification, relying instead on trivial arguments. This work does not meet the high standards of conceptual and technical novelty typically published in Nature Communications, particularly when compared to other ferroelectric FET studies. Therefore, I cannot support its publication.

Reviewer #3

(Remarks to the Author)

I am satisfied with the revisions. I would recommend acceptance of the current version of manuscript for publication.

Version 3:

Reviewer comments:

Reviewer #1

(Remarks to the Author)

It was well revised and can be accepted for publication.

Reviewer #2

(Remarks to the Author)

The authors have provided a lengthy response that lacks sufficient experimental evidence. Despite the extensive argumentation, the referee remains unconvinced due to a lack of solid scientific backing; the studies and conclusions are overly simplistic for the current stage of research on hybrid organic-inorganic perovskites. At this point in the field's development, the findings will not significantly benefit the scientific community. As a result, the referee does not support acceptance.

Merely demonstrating the existence of ferroelectricity at the single-unit cell level for hybrid organic-inorganic perovskites is inadequate. It is crucial to show that monolayer structures perform better in terms of switching or polarization compared to thicker films. Given the semiconductor/insulating nature of these systems, it is expected that ferroelectricity can be maintained, especially when insulative organic cations are present, as the depolarization effect is notably reduced. Numerous studies have already showcased ferroelectricity at the monolayer level in 2D materials.

Switching metrics and polarization can be effectively compared within the same measurement system by varying thickness. Cross-system comparisons can also be made using the same frequency or thickness.

Ultimately, the referee defers to the editor for a final decision based on the provided comments.

Response to Reviewers' Comments

(Referee comments in black; Author responses in blue; revised parts of manuscript in red)

Reviewer #1 (Remarks to the Author):

In this manuscript, the authors reported an organic-inorganic hybrid perovskite nanoflake with switchable polarization, a narrow bandgap and robust photoelectric properties down to the monolayer limit. Building on previous work, they further explored the impact of the size of ferroelectric thin films on the photoelectric effect and investigated the underlying mechanisms. Their research provides certain insights into the photoelectric applications of ferroelectric thin films. I suggest that the following minor modifications should be made before it can be published.

Response: We thank you for the positive comments and acknowledgment of the significance of our research.

1 In the title “A van der Waals hybrid perovskite ferroelectric semiconductor”, what is the “van der Waals hybrid perovskite”. Here, in the Introduction, it should be introduced, and the introduction should provide the characteristics of this structure and similar structures reported. In this way, we can understand why you mentioned this word in such a prominent title.

Response: Certainly. We have revised the introduction to include additional background on Van der Waals (VDW) hybrid perovskites, information about this material, and more examples of hybrid perovskite ferroelectrics (HPFs) with similar structures. Here we use the term “VDW hybrid perovskite” to distinguish between VDW-bonded and non-exfoliatable layered HPFs structures (see Fig. R1 for examples of comparison).

[Figure redacted]

Fig. R1. Crystal structures of (a) PEPI and (b) (R-3AMP)PbBr₄ [Adv. Mater. 34, no. 51 (2022): 2204119]. PEPI is an example of HPFs with a layered structure and VDW-bonded organic components, while (R-3AMP)PbBr₄ illustrates the typical structure of HPFs composed of layered inorganic components without any VDW bonding in the lattice. Their organic components are generally linked directly to the inorganic layers by much stronger hydrogen bonds (e.g., NH-Br bonding in (R-3AMP)PbBr₄).

2 In Fig. 1d, what does each line represent? Please label them.

Response: We note and apologize for the confusion. They are ferroelectric hysteresis loops obtained under different electric fields. We have revised the figure accordingly.

3 In the DSC results (Fig. S3c), why are there two small peaks at 120°C and 50°C during the heating process? And it's best to use Kelvin temperature for all. Different types of samples were made to compare with Ref.21, Is there any impact or difference? Because the microscopic crystalline states are actually the same.

Response: We think that the small peaks shall be caused by solution residues and maybe small signal vibration. As it can be easily observed, these peaks only appear during heating, suggesting that they shall not be the result of intrinsic thermal behaviors such as phase transitions, since phase changes are typically reversible. To verify this, we performed DSC measurements after a quick heating and cooling cycle within the same temperature range. As shown in Fig. R2, the crystal undergoes two reversible phase transitions, consistent with the previous report (Ref. 21, now Ref. 27 in revised manuscript). Obviously, the small peaks disappeared. We have revised the DSC figure with the new data and converted the temperature scale to Kelvin. And yes, the crystal is

identical to that in Ref. 21. This measurement serves as confirmation of its composition and phase transitions.

Fig. R2. DCS measurement of PEPI crystal after a quick heating and cooling cycle within the same temperature range.

4 In Fig. S3b, why is the line in the PXRD result from experimental testing not flat? It is recommended to show in normalization.

Response: The background is generated by the glass slide substrate and appears in all our XRD measurements. We have not attempted to fully remove it, as we believe the raw data shall be conveyed to the readers. Moreover, the overall background intensity is much lower than those of the main peaks and should not affect the interpretation of the XRD spectra. And of course, we have normalized the units following your suggestion (see Fig.S3b).

5 You mention “PEPI thin films were first exfoliated from bulk crystal by Scotch tape”, which crystal plane did you use to obtain the film? The carrier mobility may differ for different crystal planes.

Response: Sorry for the confusion. Similar to other 2D materials, we found that the thin film can only be obtained by attaching tape to the *bc* plane (see the coordinates in Fig. 1a; it is the plane of the 2D inorganic layers) of the bulk crystal and exfoliating it along the out-of-plane direction (the *a*-axis in Fig. 1a). The layered structure is bonded by VDW forces only in this out-of-plane

direction. Exfoliating in any other direction results in crystal cracking similar to a 3D structure, as this process involves the breaking of covalent bonds. The photoelectric and electrical measurements were conducted along the polarization direction.

We have revised the methods section to include specifics about the exfoliation direction.

6 finally, you mentioned that polarization switching in (IPA)₂(EA)₂Pb₃I₁₀ flakes with different thicknesses. First, I want to know the mechanism behind the difference in thickness, is it for conduction? Or a weak interaction? Or something else. Second, what is the impact of electric field action.

Response: We understand that the Reviewer is interrogating the mechanism of a potential size effect generated by lowering the thickness in the monolayer regime. Therefore, we will discuss the origin of ferroelectricity in PEPI flakes and explain the reasons for conducting the thickness-dependent study.

The primary reason for us to conduct the thickness (number of layers)-dependent PFM study is to confirm the intrinsic ferroelectricity of HPFs, which means the polarization is independent of the number of layers in the flake system. This distinguishes HPFs from many 2D inorganic ferroelectrics that exhibit thickness-dependent ferroelectricity, where ferroelectricity arises from specific stacking or interlayer effects and exists only in flakes with a certain number of layers (thickness) [Nature Materials, 22(5), 542-552]. For example, boron nitride (BN) exhibits ferroelectricity only in a bilayer system with "AB" stacking, while its bulk form or flakes with different numbers of layers are non-polar [Science, 372(6549), 1458-1462]. The present thickness-dependent study confirmed that the ferroelectricity of HPFs does not exhibit this thickness dependence due to unexpected effects. At a more general level, we think that conducting a thickness-dependent PFM study is a good practice for investigating the ferroelectricity of VDW materials, as it allows us to observe potential size effects.

Similar to other HPFs systems, the positive charges of the organic cations (IPA⁺ and EA⁺ in this case) come from their ammonium ends. For ease of polarization analysis [Nature Materials, 20(5), 612-617], the position of the nitrogen (N) atom is generally regarded as the center of each cation, while the position of the metal atom (Pb in this case) is considered the center of each metal-halide anion (PbI₆²⁻ octahedra; see the molecular structure in Fig. R3a). The polar structure of PEPI in each IPA-EAPbI-IPA unit layer can thus be identified as shown in Fig. R3b, which illustrates the structure with polarization oriented to the left. For each pair of adjacent cations and anions, the vertical projection of the dipole pair offsets each other, while their projection along the horizontal direction remains, resulting in long-range ordered electric polarization along the in-plane direction. This intrinsic origin of ferroelectricity indicates that the spontaneous polarization of HPFs is in principle independent of the thickness of the crystal (Fig. R3c), as thickness does not influence the structure or molecular arrangement of each layer unit.

Fig. R3. Polarization mechanism of PEPI. a, The structures of the cations (IPA⁺ and EA⁺) and anion (PbI₆²⁻). The N, Pb atoms and ammonium molecules are highlighted by red circles, orange circle and blue triangles, respectively. b, The structure and polarization analysis of a single layer unit of PEPI with leftward polarization. c, The structure and polarization of a multilayer PEPI system with leftward polarization.

Regarding the second part of your question about the impact of the electric field, do you mean to refer to the switching behavior? The ferroelectricity of PEPI originates from the off-center ordering of cations and anions, giving it intrinsic ferroelectricity same to that of classical ferroelectrics. When the electric field is sufficiently strong, the lattice can overcome the energy barrier and switch among equivalent polarization states. However, the action of the electric field will not induce or remove ferroelectricity, as spontaneous polarization is an intrinsic property of the lattice.

Reviewer #2(Remarks to the Author):

Referee's report on "A van der Waals hybrid perovskite ferroelectric semiconductor with 1 switchable polarization and superior photoelectric robustness down to 2 monolayer" by Yunzhong Hu et.al.

The authors reported device testing on a perovskite ferroelectric semiconductor, named (isopentylammonium)2(ethylammonium)2Pb3I10 (PEPI). They observed switchable polarization and exceptional photoelectric durability reaching down to a thickness of just two monolayers. This organic-inorganic hybrid perovskite nanoflake shows switchable polarization, a bandgap ranging from 1.86 eV to 2.21 eV (from bulk to monolayer), and photoelectric characteristics even at the monolayer scale. However none of these data can be considered as unique or new in the light of what had been published in the field. Consequently, I do not support the publication of this work in Nature Communications as the findings were very routine and there was no new physics or chemistry.

Response: We do not agree that our work should be denied because of the reasons above. Potentially, we may not have provided a comprehensive discussion in the introductory part to clearly outline the gap in the field of nanoscale ferroelectric semiconductors, and in general the hybrid perovskite ferroelectrics (HPFs) is filling. This may have caused some confusion regarding the research significance of our work. In response to these concerns, we would like to elaborate in detail.

Additionally, we are puzzled by your reference to "down to a thickness of just two monolayers." The thinnest flake we investigated in this work should be regarded as "one monolayer" rather than "two monolayers". And here is why. Similar to other van der Waals (VDW) structures, a monolayer of a given 2D material is defined as the smallest periodic layer unit bonded together by VDW forces. According to the crystallographic results, this can be identified as the IPA-EAPbI-IPA structure. The structure illustrated in Fig. 1a is a bilayer system, in which we aim to show the layered structure along with VDW bonding. A monolayer should be considered half of that. As discussed in the manuscript, the thickness of the exfoliated monolayer is confirmed by AFM characterizations and is consistent with the single-layer (IPA-EAPbI-IPA structure) thickness obtained from the crystallographic results.

This work is "opportunistic" as it attempts to exploit the interest in 2D ferroelectrics but failed to show more than trivial data. The novelty of the work may be questioned concerning its distinctiveness compared to existing published literature in the field for several reasons:

First there is rather limited novelty in material design. The use of organic-inorganic hybrid perovskite materials and the exploration of their ferroelectric and photoelectric properties have

been subjects of numerous research. The overall concept of using such hybrid materials in ferroelectric semiconductors has been previously explored.

Response: We do not agree that the significance of our research should be rejected because of existing studies on bulk HPFs, as bulk properties are not relevant to the topics or the challenges of 2D ferroelectric semiconductors that we aim to address in this study.

It is worth noticing that all previous studies focusing on HPF semiconductor properties are bulk crystal investigations. We believe that the significance of nanoscale-focused work should not be denied simply because of existing studies on bulk properties, as bulk and nanoscale research are obviously different fields and correspond to different applications. The bulk properties of 2D materials such as transition metal dichalcogenides (TMDs) were investigated many years ago. Yet the community does not reject their nanoscale research simply due to the existence of their bulk property studies. Based on the research significance of our work, here we explain how our work is indeed different from previous bulk property studies on HPF semiconductors.

Firstly, the present work addresses critical challenges in the field of ultrathin ferroelectric semiconductors: i) the scarcity of materials that can maintain together robust ferroelectricity and semiconductor properties at the nanoscale [Nature Communications, 12(1), 5896], and ii) the high costs and complexity of synthesizing and fabricating the discovered 2D inorganic ferroelectrics make the industrial mass production of ultrathin ferroelectric semiconductor devices currently impossible. [Nature Materials, 22(5), 542-552; Physical Chemistry Chemical Physics, 23(38), 21376-21384].

Since different material systems generally lead to unique applications with distinct advantages, investigating nanoscale properties and confirming ferroelectricity in different VDW ferroelectric families can significantly expand the spectrum and facilitate the realization of applications based on 2D ferroelectric semiconductors. Our work demonstrates that HPFs combine features of facile processing, excellent photoelectric properties, and layer-number-independent ferroelectricity at the nanoscale, which is rarely to be seen within the entire ferroelectric family. This not only largely enriches the small family of 2D ferroelectrics but also indicates that the synthesis and fabrication of ultrathin ferroelectric semiconductors can be achieved with solution processing HPFs, which has much easier synthesis and fabrication for practical device. This makes the application of low-cost nanoscale ferroelectric semiconductor devices feasible in the future, which is very challenging to be achieved with other 2D ferroelectrics such as TMDs (we will elaborate on this point in our response to the next part of your review).

Our work thus conveys a clear message and make the ferroelectric and semiconductor communities to recognize the potential of a new family of materials that includes more than seventy compositions, positioning them as ideal candidates for ultrathin ferroelectric semiconductors with the advantages of layer-number-independent ferroelectricity, great photoelectric robustness and facile synthesis/ fabrication. None of the previous studies on HPFs semiconductor have contributed

to addressing these critical challenges of 2D ferroelectric semiconductors, nor have they had a similar impact.

Secondly, our work addresses the most critical issues related to HPFs as nanoscale ferroelectrics from both experimental (the survived ferroelectricity and photoelectric properties) and theoretical perspectives (limitations of the classic Hecht equation)—issues that have never been successfully tackled in any previous studies on these specific materials and in general on semiconductor ferroelectrics. The nanoscale ferroelectric research of VDW HPFs and other 2D ferroelectrics are obviously at different stages. VDW HPFs gained significant attention around 2014 due to their exfoliable characteristics, solution synthesis and impressive photoelectric robustness [Natl. Sci. 10.2 (2023): nwac240]. Although more structures have been discovered in the past ten years, the uncertain preservation of ferroelectric and photoelectric robustness at ultrathin thickness has halted relevant nano-studies, which should be one of the most promising research fields. The confirmed properties, developed equation, and the demonstrated advantages of HPFs as ultrathin ferroelectric semiconductors position HPFs within a brand-new field of nanoscale study. This will undoubtedly accelerate related nano research.

Based on the reasons mentioned above, we believe our work significantly advances research on HPFs and ferroelectric semiconductors. These clearly distinguish our work from previous studies on HPF bulk semiconductors. Consequently, we do not agree that the significance of this research should be denied because of existing studies on HPFs bulk. They are obviously not relevant.

The narrow bandgap and robust photoelectric properties down to the monolayer level is not so significant, similar properties have been reported in other 2D materials or material systems, albeit with variations in specific values. I don't see why this is considered novel, in view of the fact that depolarization effects in hybrid perovskites is less compared to 2D inorganic system because the former is more insulating and thicker!

Response: We cannot agree that the significance of a study should be rejected simply based on a certain figure of merit (FOM) or properties, such as narrow bandgap is also available in another entirely different material systems. FOM does not equate to research significance, and realizing the same property in different material systems absolutely carries different impact. If a low-cost solution synthesis is developed and can be applied to general 2D inorganic ferroelectrics, it is impossible that the research community will reject such work simply because solution methods are already available for hybrid perovskite materials. A similar reasoning applies to the demonstration of ultrathin ferroelectric semiconductors through hybrid perovskites. Ultimately, the goal is to achieve practical 2D ferroelectric semiconductors, but use different material systems.

For hybrid perovskites, one of the primary reasons that they have received significant attention in various fields is not that they can outperform other materials in certain FOM. For their most important application solar cells, the current energy conversion efficiency is around 25%, while

classic gallium arsenide can easily exceed 45% (see <https://www.nrel.gov/pv/cell-efficiency.html>). Instead, the key factors are their tolerant and cheap solution synthesis and ease of fabrication [Nature Reviews Materials, 3(4), 1-20], whereas the synthesis of other materials such as inorganic semiconductors and ferroelectric generally relies on high-temperature chemical vapor deposition (CVD, e.g., for TMDs) or high-energy pulsed laser deposition (PLD, for traditional ferroelectric oxides). Large hybrid perovskite single crystals can be synthesized simply after obtaining the necessary chemicals and solutions, while large-area semiconductor thin film devices can be fabricated using various easy technics such as spin coating and doctor-blade. This unique feature largely reduces production costs and offers really foreseeable benefits for practical devices, consequently sparking considerable interest in the underlying basic research [Chemical Society Reviews, 49(22), 8235-8286]. When assessing the research significance of a hybrid perovskite study, if we overlook this practical value and evaluating research significance just by FOM comparison, most hybrid perovskite studies should be classified as failing to demonstrate breakthroughs in FOM and carry negligible research significance. It is obviously not the case.

For ultrathin ferroelectric semiconductors, one great challenge is the overall high cost coming from synthesis and device fabrication, which render mass industrial production based on current discovered 2D ferroelectrics impossible [Physical Chemistry Chemical Physics, 23(38), 21376-21384; Nature Materials, 22(5), 542-552]. The ferroelectricity in many of these materials only exists in flakes with a specific number of layers [Nature Materials, 22(5), 542-552]. Consequently, for device fabrication, it is generally necessary to first conduct thickness measurements/identifications, and then selectively pick films with the target thickness from all exfoliated flakes. Additionally, the expensive CVD/PLD synthesis facility of inorganic ferroelectrics and associated high temperature/energy supply largely increase the cost of product. Our work demonstrates HPFs persist the combined features of intrinsic (layer-number-independent) ferroelectricity, solutions synthesis and great photoelectric activity in ultrathin systems, which is rarely to be seen in the whole ferroelectric family. This indicates HPF nanofilms are ferroelectric and photoelectrically active regardless of thickness and are ready for use just after exfoliation or spin coating. High-performance ferroelectric semiconductor nanostructures can be available to industrial and academic professionals through cheap and convenient solution process just upon obtaining necessary chemicals. This undoubtedly significantly advances the field of practical 2D ferroelectric semiconductors. Whether there is slight improvement in FOM such as bandgap compared with all other material system is clearly unrelated to this topic nor have any influence on addressing the discussed challenge of ultrathin ferroelectric semiconductor. We believe it is similar to rejecting a work that contribute to addressing issue "A" by reason that it does not demonstrate a breakthrough in enhancing property "B".

Regarding the smaller depolarization field, the slight thickness difference between 2D inorganic ferroelectrics and HPFs shall not be relevant. The depolarization field arises from the incomplete compensation of ferroelectric polarization; thus, its direction aligns only with the polarization direction, and its magnitude is inversely proportional to the material dimension along the

polarization direction [Journal of Applied Physics, 44(8), 3379-3385]. This explains the dramatic increase of depolarization field observed in traditional ferroelectrics and ultrathin flake systems with out-of-plane polarization, as the dimension scaling and the polarization in these materials are along the same direction. As discussed in our manuscript, most 2D HPFs exhibit in-plane polarization, meaning that the direction of dimensional scaling is perpendicular to the polarization direction. HPF flakes can reach lengths of tens of micrometers in the lateral size while their thickness are at the nanometer level, resulting in significantly lower depolarization fields. This is a natural feature and advantages for HPFs in terms of depolarization field and polarization stability.

The comparison with traditional oxide ferroelectric semiconductors in terms of photoconductivity is selective and biased. Photoconductivity is a function of absorption and also conductivity of the material and it is meaningless comparing to wide band gap system.

Response: Here, we make this comparison because it is directly related to one of the main issues: the generally low photoelectric signal in traditional ferroelectric semiconductors [Science Advances, 4(7), eaat3438]. Photoconductivity is one of the most important FOM for photoelectric applications as it determines the magnitude of the photocurrent. The low conductivity, low absorption coefficient, and wide bandgap are also the reasons for the low performance of many traditional ferroelectric semiconductors at the nanoscale [Advanced Functional Materials, 32(14), 2109625]. For instance, the low absorption coefficient is a key factor that prevents many traditional ferroelectric semiconductors from going to ultrathin, as the light simply requires a long penetration depth in these materials to be fully absorbed and generate photocurrent. Photoconductivity relates to these parameters and reflects the overall photoelectric robustness and is directly relevant to the application discussed. We believe that when referring to a very specific application, we can state that factors such as low absorption are the reasons for the disadvantageous properties of a certain material system, but we cannot claim that these factors invalidate property comparisons. Therefore, we would like to keep our opinion that photoconductivity is a reasonable parameter for comparison here and serves to demonstrate the advantages of HPFs as 2D ferroelectric semiconductors.

While the use of the nanodevice Hetch equation to estimate mobility-lifetime product and photoelectric robustness is a novel aspect of the work, the overall significance of this methodology in the broader context of material characterization and analysis needs to be further established. The extent to which these findings significantly advance the field compared to existing research on similar materials or device architectures is highly questionable.

Response: We thank you for approving this part of research novelty of our work.

As we all know, a common practice for assessing the applicability of a given equation to a specific material system is to verify whether its assumptions, boundary conditions, and physical principles are consistent with each other. The classic Hecht equation is developed based on principles only

related to basic optical physics, without any assumptions or boundary condition applicable to specific material systems [Zeitschrift für Physik, 77(3), 235-245]. As a result, the equation is widely utilized across a great variety of material systems without requiring further analysis, including classic semiconductors like silicon [Applied Physics Letters, 46(4), 405-407], metal halides [Journal of Applied Physics, 91(5), 3345-3355], and non-hybrid perovskite solar cell thin films [Physica Status Solidi (A), 208(8), 1813-1816]. Compared with the classic equation, the modifications we made on the original equation are addressing actual boundary conditions of practical experiment. These changes are, obviously, not addressing the underlying physical principles, but help future investigators to properly use in usual conditions of uniform illumination, irrespective of addressed material system and thus not influence the applicability of equation.

But of course, we are happy to further validate the applicability and accuracy of our developed equation through experiments. To test the correctness of the equation, we conducted photocurrent measurements on a spin-coated MAPbBr₃ thin film, which is one of the classic hybrid perovskite structures [Advanced Electronic Materials, 8(4), 2100980], with the same device architecture and estimate $\mu\tau$ by IV curve (see Fig. R4a). The obtained value of $2.7 \times 10^{-7} \text{ cm}^2/\text{V}$ is highly consistent with those of previous reports ($\sim 10^{-7} \text{ cm}^2/\text{V}$) focusing on MAPbBr₃ polycrystalline thin films studies [ACS Energy Letters, 3(6), 1233-1240; ACS Applied Materials & Interfaces, 12(21), 24498-24504]. This validates the accuracy of our equation and demonstrates its applicability to different material forms (PEPI single crystal flakes and polycrystalline MAPbBr₃ thin films). Next, we conducted the same characterization on PM6:Y6 thin films, a well-known pure organic photovoltaic material [Advanced Materials, 36(20), 2302005]. The obtained $\mu\tau$ value of $8.81 \times 10^{-9} \text{ cm}^2/\text{V}$ (see Fig. R4b) also shows strong consistency with previous reports of $\sim 10^{-9} \text{ cm}^2/\text{V}$ [Advanced Energy Materials, 11(22), 2100804; Advanced Materials, 35(9), 2210463]. This demonstrates that the developed equation can be employed across different material systems. We have included these results in our revised supporting information.

Fig. R4. I-V characteristics of (a) MAPbBr₃ and (b) PM6:Y6 thin films using the same device structure as the PEPI flake for $\mu\tau$ estimation.

Overall, we cannot agree that our work should be rejected for the reasons you mentioned. Of course, we also acknowledge that we may not have provided a comprehensive introduction, which may lead to some misunderstandings. We have made revisions to the manuscript to clearly outline the challenges associated with 2D ferroelectric semiconductors and explain how HPFs can address this gap.

Reviewer #3 (Remarks to the Author):

Through the exfoliation of PEPI crystal, a van der Waal hybrid perovskite ferroelectric semiconductor, the authors successfully obtained thin nanoflakes that survive the ferroelectricity even down to the monolayer limit. Moreover, the authors demonstrated that the PEPI monolayer exhibits superior and robust photoelectric properties, with photoconductivity several orders higher than that of traditional oxide ferroelectric semiconductors. The manuscript demonstrates the promising potential of PEPI nanoflakes in the realm of nano-photoelectronic devices, illuminating new avenues for future optoelectronic applications. This study, therefore, presents some novelty in the field. After addressing the following comments, this work could be accepted for publication.

Response: We thank you for your positive comments and approval regarding the novelty of our research.

1.The inclusion of pioneering literature on Van der Waal hybrid organic-inorganic ferroelectric semiconductors is essential for a comprehensive understanding (Nat Commun 6, 7338 (2015)).

Response: We thank the Reviewer for this valuable suggestion. We have cited the relevant work (see ref.23).

2.For the verification of the ferroelectricity in PEPI nanoflakes, especially for the monolayer sample, the PFM signal is too weak to judge a definite polarization reversal. The in-plane PFM signal correlates with the component of in-plane polarization projected onto the cantilever normal. Thus, it is suggested to reorient the sample orientation with respect to the PFM cantilever to enhance signal strength. In addition, given such a weak PFM response, the authors are advised to distinguish between contributions from ferroelectric switching and electrostatic influences.

Response: We fully agree with the sample reorientation strategy for PFM characterization. Orienting the polarization to be perpendicular to the cantilever is indeed the method we employed during scanning. However, a decrease in piezoelectric signal with thickness scaling is sometimes inevitable in various 2D systems [Nature Communications, 7(1), 1-6; Applied Physics Letters, 86(1)]. Although the piezoelectric signal is noticeably weaker in thinner flakes, we believe the present results should be sufficient to demonstrate the switchable ferroelectricity in PEPI thin films, as clear piezoelectric and switching behavior can be distinguished from potential electrostatic influences.

Specifically, as shown in Fig. 1b, the regions on either side of the sweeping track were switched to different directions, with the domain boundary aligning with the sweeping line. Electrostatic influence can surely play a role during scanning, yet the observed switching feature—where both sides are switched to different polarization directions—is highly unlikely to be caused by electrostatic effects. Consequently, the switching features related to: 1) the two sides of the

sweeping track switching to different directions and 2) the domain boundary aligning closely with the sweeping track demonstrate that the PFM signal is primarily attributed to piezoelectricity rather than other effects such as electrostatic forces.

Of course, it is good practice to run sanity check experiments and, in this particular case, reaffirm the piezoelectric origin of the PFM signal. To this end, we conducted a temperature-dependent PFM study. Fig. R5 shows the PFM results of a crystal flake at room temperature and at 343 K (PEPI transitions to the paraelectric phase at 340 K). The flake exhibits an obvious multidomain structure and decent amplitude at room temperature; however, both features disappear after the phase transition. The strong correlation between the PFM signal and the phase transition provides another solid evidence that the PFM signal in our characterization primarily originates from piezoelectricity rather than electrostatic signals or other effects. We have included the temperature-dependent PFM measurements in our revised supporting information.

Fig. R5. Temperature-dependent PFM study of a 400 nm PEPI film at room temperature (ferroelectric phase) and at 343 K (paraelectric phase). The significant reduction in amplitude and the disappearance of the multidomain feature at elevated temperatures demonstrate the piezoelectric origin of the PFM signal.

3. It is suggested to incorporate corresponding color scales to Fig. 2b and c, as well as Supplementary Fig. 5 to 7, to enhance the interpretability of the data presented.

Response: We have made this revision following your suggestion. Please see Fig. 2b-c and Supplementary Fig. 5-7.

4. What are the measurement conditions for the different curves in Fig. 1d and the inset of Fig 3b.

Response: The curves in Fig. 1d are ferroelectric hysteresis loops obtained under different electric fields. We have labeled all these loops by corresponding electric field in the revised manuscript. Fig. 3b and its inset show the IV curves of the same device under focused and defocused laser illumination. The main difference between these two is that, in the former case, the illuminated area covers only a very small portion of the sample surface, whereas in the latter, the illumination covers the entire crystal surface. This results in a higher photocurrent due to the larger illuminated area and shorter diffusion length for photoexcited carriers near the electrode (the illumination of defocused laser can cover this part while focused laser illumination cannot). To avoid confusion, we have revised the diagram and added more explanations in the caption. The IV curves obtained under the same laser power in Fig. 3b and inset are now labeled with the same color.

5. The polarization of PEPI is along the in-plane direction. However, there exists an offset along the out-of-plane direction between dark and photocurrent curves, as illustrated in Fig. 3c. It means that the device has a current response in the out-of-plane off state at zero bias. I wonder what causes this offset and whether it is associated with the polarization direction.

Response: We thank the reviewer for flagging us this. The small current was an artefact of the measurement. It is due to the large relaxation time in the very low current regime doubled by the long integration time of the source-meter. We have performed new measurements with optimized contact as well as the instrument integration time and the offset is negligible now (see Fig. R6). We have updated the figure in manuscript accordingly.

Fig. R6. Comparison between the photocurrent and dark current density along the in- and out-of-plane directions

Response to Reviewers' Comments

(Referee comments in black; Author responses in blue; revised parts of manuscript in red)

Reviewer #1 (Remarks to the Author):

It was well revised and can be accepted for publication.

Response: We thank you for the help in improving the manuscript and your approval of the research significance of our work.

1. The authors should clarify the orientation of the sample when studying the J-V curve, as ferroelectric materials possess the ability to self-drive.

Response: Apologies for the confusion. We have added orientation information to the relevant sections of the main text and methods parts. Please refer to the highlighted parts of our manuscript.

2. Can the authors provide the butterfly curve during PFM measurements? Besides the switching of domain walls, this data is also very important.

Response: We would like to clarify that our material exhibits in-plane polarization. Additionally, we would like to emphasize that PFM measurements are not helpful or necessary for demonstrating the in-plane ferroelectricity. The clear demonstration of ferroelectricity is illustrated by the classical ferroelectric hysteresis loop shown in Fig. 1d of the main text, distinct polarization in pristine nanoflakes (Fig. 2c), consistent switching behaviors (Fig. 2b), and the temperature-dependent PFM images showing correspondence between ferro-paraelectric phase transition and PFM signal (Supplementary Figs. 5-7). The PFM butterfly loop is a typical feature of a PFM measurement with an out-of-plane geometry used to characterize a ferroelectric material with a significant component of the ferroelectric polarization pointing perpendicular to the sample surface sampled by the AFM tip. The in-plane PFM hysteresis loops show only the local shear displacement vs. applied field. Consequently, these in-plane PFM loops should not produce any “butterfly” loops, as the amplitude of shear displacement under the AFM tip is not univocally related to the dynamic processes of local switching. The sample surface displacement detected by AFM tip is a complex 3D move given by the tensorial relation:

$$x_i = d_{ij}E_j$$

The shear strain detected by in-plane piezoresponse force microscopy (PFM) is influenced solely by the non-diagonal components of the piezoelectric tensor d_{ij} , which are not necessarily dependent on the direction of polarization. Additionally, when a DC bias is applied, the in-plane polarization is switched due to the stray field generated under the tip, allowing a domain wall to form directly beneath it. In contrast to the out-of-plane scenario, the amplitude of the in-plane PFM signal after switching, which results from the shear movement of both domain orientations, is very low, nearly zero. The piezoelectric effect and shear movement induced by material at the different sides of domain wall (tip) can be with similar magnitude but opposite directions. This is the primary reason why in-plane PFM amplitude-voltage loops are seldom reported in high-profile journals.

In summary, we believe that this specific data is not unimportant and can be detrimental to the overall quality of the paper.

3. The author can replace Table 1 with graphics to better demonstrate the advantages of the nominated compounds compared to other reported materials.

Response: Thank you for your valuable suggestion. We have included the bandgap data in our revised Fig. 4f (see Fig. R1 below). We noticed that some of the information in the table, such as synthesis method and thickness, are about comparisons regarding the synthesis cost and exfoliation ability of the materials, which are not quite suitable for conversion into a graph. The photocurrent density here is highly relevant to the photoconductivity shown in Fig. 4f. Therefore, we would like to add the bandgap information to Fig. 4f while keeping Table 1 for reference regarding the other information.

Fig. R1. Photoconductivity and bandgap (E_g) comparisons of different ferroelectric materials.

Response to Reviewers' Comments

(Referee comments in black; Author responses in blue; revised parts of manuscript in red)

Reviewer #1 (Remarks to the Author):

It was well revised and can be accepted for publication.

Response: We thank you for the supportive feedback regarding Reviewer 2's comments and your help in improving the manuscript!

1. The authors should clarify the orientation of the sample when studying the J-V curve, as ferroelectric materials possess the ability to self-drive.

Response: Apologies for the confusion. We have added orientation information to the relevant sections of the main text and methods parts. Please refer to the highlighted parts of our manuscript.

2. Can the authors provide the butterfly curve during PFM measurements? Besides the switching of domain walls, this data is also very important.

Response: We would like to clarify that our material exhibits in-plane polarization. Additionally, we would like to emphasize that we have demonstrated in-plane polarization switching through PFM imaging, while PFM switching spectroscopy (butterfly curve) will not provide additional information for the in-plane polarization switching, as explained in the following text.

Specifically, the butterfly loops observed in the out-of-plane PFM measurements indicate the amplitude and phase changes during polarization reversal under a sweeping electric field (E). Here, the amplitude signal is measured along the out-of-plane direction, due to the longitudinal piezoelectric effect induced by the applied small AC bias superimposed on the bipolar electric field. This amplitude is thus positively correlated with the magnitude of the net polarization [Nature Communications, 10(1), 1661]. Fig. R1 illustrates the applied switching bias, butterfly loop, and the states of the PFM tip/polarization during this switching. The moments 0 to 4 in Fig. R1a indicate the following states: moment 0 corresponds to the pristine single-domain state of the material, moment 1 represents the state when the electric field is at positive coercivity, moment 2 is at the positive maximum, moment 0' is at zero again, moment 3 is at negative coercivity, and moment 4 corresponds to the negative maximum. From 0 to 1, the amplitude firstly decreases due to the reduction in net polarization resulting from switching (see Fig. R1b and c). From 1 to 2, the amplitude increases as the E will fully switch and stretches the polarization, leading to larger piezoelectric effect. Then from 2 to 3, the amplitude decreases again as the stretched polarization first returns to its pristine magnitude (from 3 to 0') and then is partially switched to another

direction (from 0' to 3). The amplitude changes from moments 3 to 4 and back to 0 resemble the changes from 1 to 2 and to 0', but with the polarization change in the opposite direction. This sequential change in amplitude accounts for the butterfly loop observed in out-of-plane PFM measurements. The electric field at the amplitude minima (point 1 and 3) represents the coercive field required for polarization switching.

Fig. R1. Applied bipolar E (a), butterfly loop (b) and cantilever/sample polarization states (c) during polarization reversal of out-of-plane PFM measurement.

The amplitude-bias curve obtained from the in-plane PFM measurements is quite different, which records the cantilever torsional motion due to the shear piezoelectric effect (Fig. R2a). However, unlike in the out-of-plane PFM measurements—where local polarization can be switched between upward and downward single-domain states—the polarization on different sides of the cantilever will be switched to opposite directions in the in-plane amplitude-voltage curve measurement, where polarization switching is realized due to the in-plane component of the radical electric field from the tip (see Fig. R2b). This means that the tip will always be at the domain boundary, resulting in zero net polarization and low amplitude as the piezoelectricity induced torsion from the left and right sides will offset each other. In practical case, this amplitude will be quite small and likely be dominated by other parasitic effects, thus carrying meaningless information. Consequently, local amplitude-bias curve measurements are rarely seen in high-profile journals, as they are not considered as an effective method for demonstrating in-plane polarization switching. Specifically, during the bipolar switching, as shown in Fig. R2c, the tip can have a decent amplitude at the pristine single-domain state due to the lateral piezoelectric effect. Yet from moments 0 to 2, the amplitude will decrease rapidly to zero as the left-side would be switched to another direction under the radical electric field, leading to a zero net polarization at the tip's position. After moment 0', both sides of the domain switch to directions that are opposite to their previous states due to the inversion of the radical electric field from the tip. However, this does not change the state of the tip, as it remains at the domain boundary with zero net polarization. Therefore, the amplitude

signal will be quite weak during this bipolar switching, and the dynamic polarization reversal on the left and right sides of the cantilever cannot be reflected by the PFM amplitude.

Fig. R2. Cantilever motion (a), switched polarization under downward (upper panel) and upward (lower panel) radical electric field (b) and cantilever/sample polarization states (c) during polarization reversal of in-plane PFM amplitude-voltage measurement.

For in-plane ferroelectrics, a much more effective approach to demonstrating in-plane ferroelectricity is conducting PFM scanning across the surface before and after line/point switching (as shown in Fig. 2b and see more examples from other works in Fig. R3). This is because the features of 1) the two sides of the sweeping track switch to different directions, and 2) the domain boundary aligns closely with the sweeping track, are highly unlikely to result from effects other than ferroelectricity.

[Figure redacted]

Fig. R3. In-plane PFM images of DIPAB (a and b [Advanced Materials, 27(47), 7832-7838]) and BiFeO₃ (c and d [Nature nanotechnology, 13(10), 947-952]) before (a and c) and after (b and d) switching.

In addition, we believe that the ferroelectricity of our nanoflake has been confirmed through multiple consistent measurements, i.e., demonstration of distinct polarization states (domain structures) in the pristine nanoflakes (PFM diagram in Fig. 2c), conventional ferroelectric P-E loops, PFM imaging of domain switching after voltage applications (Fig. 2b and Supplementary Figs. 5-7), and the temperature-dependent PFM showing correspondence between phase transition and PFM signal (Supplementary Figs. 5-7), which serves as a further validation. We hope you find our detailed explanation satisfactory.

3. The author can replace Table 1 with graphics to better demonstrate the advantages of the nominated compounds compared to other reported materials.

Response: Thank you for your valuable suggestion. We have included the bandgap data in our revised Fig. 4f (see Fig. R4 below). We noticed that some of the information in the table, such as synthesis method and thickness, are about comparisons regarding the synthesis cost and exfoliation ability of the materials, which are not quite suitable for conversion into a graph. The photocurrent density here is highly relevant to the photoconductivity shown in Fig. 4f. Therefore, we would like to add the bandgap information to Fig. 4f while keeping Table 1 for reference regarding the other information.

Fig. R4. Photoconductivity and bandgap (E_G) comparisons of different ferroelectric materials.

Reviewer #2 (Remarks to the Author):

The authors have outlined their reasons for believing that demonstrating the ferroelectricity of single-layer perovskites is significant. However, I argue that this is not particularly interesting, as it is unsurprising that ferroelectricity can persist in thin layers of such dielectric confined systems. For instance, even in purely inorganic materials like In_2Se_3 , ferroelectricity has been shown to survive down to the monolayer level. Ferroelectricity down to 1 nm has been demonstrated in Bismuth oxide. *Science*, (2023) Vol 379, Issue 663. Why should it be so special that a hybrid organic inorganic system that is easy to exfoliate will show single unit cell ferroelectricity? how is this useful ? This is not a film but a flake.

Response: We would like to emphasize that the demonstrated ferroelectric and photoelectric robustness in ultrathin hybrid perovskite ferroelectrics (HPFs) is significant not only because it enriches the ultrathin ferroelectrics family with scarcity, but also because these properties can be key to addressing the current issues faced by 2D ferroelectric semiconductor materials. The research significance of our work is merely just about fulfilling the concept of “monolayer” with “hybrid perovskites.” Therefore, we believe it should not be judged simply on whether monolayer ferroelectricity has been realized in other material systems. As the deterioration of ferroelectricity at the nanoscale is a well-recognized phenomenon for most ferroelectrics [*Advanced Materials*, 33(13), 2005098], and the importance of ultrathin ferroelectrics is easily noted from the substantial works on 2D ferroelectrics [*Nature Reviews Materials*, 8(1), 25-40]. We understand that you are questioning the importance of HPF nanoflakes as a new ultrathin ferroelectric material. Here, we provide specific explanations.

Specifically, the demonstrated combined features of HPFs nanoflake are rarely seen in other 2D ferroelectrics and can be key to addressing the main challenges faced by ferroelectric semiconductors at the nanoscale. These intrinsic features include: 1) the presence of photoelectric-robust metal-halide polyhedra, paired with demonstrated orders higher photoconductivity; 2) solution-processing capabilities; and 3) a van der Waals structure coupled with layer-number-independent ferroelectricity down to the monolayer level. These combined features hold great promise in tackling the current challenges associated with practical ferroelectric semiconductors at the nanoscale, including: a) generally low photoelectric signals in ultrathin systems [*Science Advances*, 4(7), eaat3438]; b) impractical industrial production due to high synthesis and fabrication costs [*Physical Chemistry Chemical Physics*, 23(38), 21376-21384; *Nature Materials*, 22(5), 542-552]; and c) the scarcity of available 2D ferroelectrics and the challenging device fabrication processes arising from layer-number-dependent ferroelectricity in many 2D inorganic materials [*Nature Materials*, 22(5), 542-552].

We assume you acknowledge the obvious significance of achieving photoelectric robustness, as you also mentioned the issues of large bandgap, low conductivity, and limited photoconductivity in classic ferroelectrics in previous comment. Here, we would like to further explain the

importance of realizing low-cost, easily processed 2D ferroelectric semiconductors. Our work demonstrates that ferroelectricity in the ultrathin HPF can: a) survive unexpected dimensional effects, such as interface strain observed in classic ferroelectric oxides [Nature Communications, 7(1), 1-6]; and b) be intrinsically independent of the number of layers. This contrasts with many 2D inorganic materials, where ferroelectricity is present only in flakes of mono- or bilayer thickness [Nature Materials, 22(5), 542-552], which means all exfoliated HPF nanoflakes and future spin-coated thin films shall all in principle exhibit ferroelectricity regardless of sample thickness at the nanoscale. This allows direct device fabrication without concerns about the presence of ferroelectricity or the consequent impracticalities fabrication for mass industrial production. Challenging fabrication processes and high-cost synthesis present significant barriers to the commercialization of ultrathin ferroelectric semiconductors [Physical Chemistry Chemical Physics, 23(38), 21376-21384; Nature Materials, 22(5), 542-552]. For many discovered 2D inorganic materials that exhibit ferroelectricity only at specific thicknesses, it is clearly impractical for industrial professionals to go through the challenging and time-consuming processes of identifying the thickness of flakes on a substrate, selecting flakes with the desired thickness, and transferring the chosen flakes to electrodes for each semiconductor device. Let alone the high costs associated with PLD or CVD facility and energy supply. The demonstrated thickness-independent ferroelectricity, combined with solution synthesis—which incurs negligible costs compared to the expenses associated with facilities, high temperatures, and energy requirements of traditional and 2D inorganic ferroelectrics synthesis—highlights the practical and straightforward processing characteristics of HPFs. These absolute advantages in cost effectiveness and easy processing features shall not be denied. They are crucial for overcoming the high costs associated with material synthesis and fabrication of nanoscale ferroelectric semiconductors. Practical advantages like these are not directly reflected by figures of merit (FOMs), yet they are undeniably important for realizing future applications.

We may use an imperfect but straightforward example of superconductive materials, which are generally achieved under high-pressure conditions [Nature, 569(7757), 528-531]. Some of these materials may exhibit impressive FOMs and intriguing concepts; however, the requirement for high pressure in devices significantly complicates their design, fabrication and cost, thereby hindering numerous practical applications. If a material system demonstrates relevant properties with similar, or even lower FOMs, but offers clear advantages in addressing synthesis and fabrication challenges, why should its significance be denied? It is this demonstrated work in new material systems, rather than the advancing in FOMs, that ultimately contributes to the realization of practical applications.

Our work demonstrates that nanoscale ferroelectric semiconductors can be achieved using a much more practical and cost-effective material system, which also exhibits superior ferroelectric and photoelectric robustness. This holds significant importance and is directly related to the major challenges associated with ferroelectric semiconductors at the nanoscale. We are puzzled by the perception that these demonstrated properties are unimportant and that related practical issues are

overlooked, while perhaps excessive attention is paid to whether the concept of monolayer ferroelectricity has been realized or whether improved FOMs, such as switching metrics or narrower bandgap values, have been claimed. FOMs such as low coercivity are desirable also because smaller switching voltages require less energy and cost for operation in the final system, thereby enhancing their practicality for real applications. The properties demonstrated by HPF flakes and the messages conveyed in our work are far more significant than merely fulfilling the simplistic concepts of monolayer ferroelectrics or achieving general improvements in FOMs such as bandgap values.

In summary, the research significance of our work clearly is not merely fulfilling the concept of monolayer ferroelectricity. It is irrelevant whether ultrathin ferroelectricity has been reported in other material systems, and we believe our work should not be dismissed on this basis. Even from the perspective of enhancing FOM, we have demonstrated orders higher photoconductivity, which is one of the most important FOM for photoelectric transistors, in ultrathin HPFs compared to classic ultrathin ferroelectric semiconductors. The developed nanoscale Hetch equation also has wide applicability for 2D photoelectric systems. These shall carry undoubtable research significance. We hope we have provided a clear explanation and addressed your concerns.

Additionally, the claim that hybrid perovskites differ based on synthesis costs is not particularly relevant and comes off as a trivial excuse. Simply reducing the material to thinner layers without any improvement in switching metrics compared to bulk—merely stating that polarization and coercive fields are similar to previous reports—demonstrates a lack of understanding.

Response: We are puzzled by the statement that successful demonstrations of properties in more practical and low-cost material systems are irrelevant to research significance or should be dismissed as mere unimportant excuses. In most research fields can possess application value, investigating and demonstrating relevant physical and chemical properties in material systems that with lower costs and easier processing feature is undeniably recognized as highly important and is generally considered common sense. This is one of the reasons there are large communities focusing on organic and hybrid perovskites across diverse fields, including solar cells, ferroelectrics, and electronic devices. As applications are inherently related to commercialization and practicality, how is it possible that material investigations value only advancements in FOMs or intriguing concepts while disregarding the significance of cost considerations such as property demonstrations in much lower-cost material systems? We really do not understand this perspective, and our community certainly does not deny the importance of low cost and easy processing features of these material systems.

We are also puzzled by the claim that the lack of improvement in switching metrics, such as coercivity at the nanoscale, indicates our lack of understanding. Firstly, depending on the specific case and mechanism, the coercivity of ferroelectrics can either increase [Nature, 391(6670), 874-877], decrease [ACS Nano, 12(5), 4736-4743], or remain relatively independent [Applied Physics

Letters, 89(23)] with dimensional scaling. It is incorrect to assume that coercivity will always decrease, and this change is not indicative of the researchers' understanding. Secondly, we have neither attempted nor claimed that coercivity will be greater or smaller with dimensional scaling. The polarization of our nanoflakes is switched by a quasistatic radial electric field during PFM measurements, whereas the polarization in previous reports is switched by a 25 Hz AC electric field oriented parallel to the polarization direction [Nature Communications, 12(1), 284]. It is common knowledge that ferroelectric coercivity depends on electric field frequency, making it scientifically incorrect to extract coercivity from—or compare the magnitudes of—electric fields with different directions and forms. Consequently, the assertion that our understanding is limited due to the absence of an improvement in switching metrics is unreasonable.

Furthermore, the compatibility of in-plane ferroelectricity with the two-terminal vertical devices required for ferroelectrics is not addressed, suggesting that this work is out of touch with the current field and may not benefit the community. I do not find any new insights in this study, and the authors' responses have not provided substantial justification, relying instead on trivial arguments. This work does not meet the high standards of conceptual and technical novelty typically published in Nature Communications, particularly when compared to other ferroelectric FET studies. Therefore, I cannot support its publication.

Response: We do not agree with the assertion that our work is out of touch with the current field simply because ferroelectricity has not been demonstrated in a particular device structure, such as two-terminal vertical devices. Our study focuses on materials science with characterizations mainly in horizontal systems, and science-oriented research obviously does not involve comprehensive device investigations based on all device structures. The lack of detail device-oriented investigations is common in materials science studies that focus on demonstrating ultrathin ferroelectrics, including the works on bismuth oxide [Science, 379(6638), 1218-1224] and In₂Se₃ [Science Advances, 4(7), eaar7720], which you also referred to. Such demonstrations are essentially beyond the scope of research aimed at investigating the ferroelectric and photoelectric behaviors of a new material system at the nanoscale.

More importantly, we believe it is improper to assess the compatibility of a ferroelectric material with the current field solely based on its ability to realize advanced devices using a specific structure. It is widely recognized that both horizontal and vertical two-terminal structures are mainstream device configurations at the nanoscale for ferroelectric semiconductors, each serving different applications [Journal of Applied Physics, 100(5); Nature Reviews Materials, 2(2), 1-14]. To achieve enhanced ferroelectric behavior in 2D HPFs with in-plane polarization, it is obvious that the electrode direction is preferred to align with the in-plane polarization direction, which typically requires the use of a horizontal rather than a vertical two-terminal system. Therefore, it is not reasonable to claim that an in-plane ferroelectrics is out of touch with the current field simply because no vertical two-terminal device has been demonstrated. Horizontal nanodevices have

numerous transistor applications and correspond to the classic CMOS FET architecture (complementary metal-oxide-semiconductor field-effect transistors) [Nature Reviews Materials, 2(2), 1-14; Advanced Materials, 33(12), 2005620]. For this reason, a significant number of both novel and traditional ferroelectrics are being investigated in the context of ferroelectric FETs and semiconductor-related research using horizontal two-terminal structures at the nanoscale [Nature Reviews Materials, 2(2), 1-14]. Recent notable studies include investigations on α -In₂Se₃ [Nature Electronics, 2(12), 580-586], MoS₂ [Nature Electronics, 7(1), 29-38], and SnTe [Science, 353(6296), 274-278].

Consequently, based on the following facts: 1) Almost no material can be guaranteed to achieve high performance across all device structures, nor is it necessary for it to do so; 2) the application value of a ferroelectric semiconductor material is not solely linked to its ability to achieve high performance in vertical two-terminal systems; and 3) the materials science-oriented nature of our work, we would like to emphasize that the assertion that our work is out of touch with the current field is unreasonable. Additionally, we would like to say that it is not reasonable to evaluate our work using the standards applied to device- or FET-oriented studies. Our work clearly focuses on investigating the ferroelectric and photoelectric behaviors of a new material system at the nanoscale, demonstrating its potential from a materials science perspective rather than from a device performance viewpoint. This approach is similar to the studies on bismuth oxide [Science, 379(6638), 1218-1224] and In₂Se₃ [Science Advances, 4(7), eaar7720] that you mentioned. Therefore, we cannot agree that our work should be dismissed for these reasons.

Nonetheless, we are happy to share some data from our ongoing device work to indicate its promise as nanoscale ferroelectric devices. In a horizontal two-terminal system, switchable bulk photovoltaic current with orders higher photocurrent density than BFO [Advanced Materials, 22(15), 1763] can be observed in PEPI nanoflakes under polarized light (see Fig. R5). The angular dependence indicates that this effect originates from bulk photovoltaic rather than other trivial effects. However, we prefer to include these results only in the response letter for the reasons mentioned earlier: specific device investigations or demonstrations of one particular structure fall outside the scope of materials science-oriented work that focuses on investigating the ferroelectric and photoelectric behaviors in an unexplored material system at the nanoscale.

Fig. R5. (a) Switchable bulk photovoltaic current in a PEPI nanoflake with a horizontal two terminal device structure under polarized laser illumination. (b) Angular dependence of the bulk photovoltaic current.

In addition, we need to say that it is not quite fair to claim that we have only provided trivial arguments in our last response letter. Through experiments and detailed explanations, we have addressed your questions regarding why previous reports on bulk HPF ferroelectric semiconductors are not relevant to the significance of our research, the true reason for the difference in depolarization fields between HPFs and other 2D ferroelectrics, the validation of our photoconductivity comparison, and the applicability of our nanoscale Hetch equation, etc. We are surprised and cannot say it is very fair to continue dismissing our work after we have addressed all these questions.

We hope you find the experiments and explanations we provided in our response letter satisfactory.

Reviewer #3 (Remarks to the Author):

I am satisfied with the revisions. I would recommend acceptance of the current version of manuscript for publication.

Response: We thank you for your approval and help in improving the manuscript!

Response to Reviewers' Comments

(Referee comments in black; Author responses in blue)

Reviewer #1 (Remarks to the Author):

It was well revised and can be accepted for publication.

Response: We thank you for all your help throughout the revision process!

Reviewer #2 (Remarks to the Author):

The authors have provided a lengthy response that lacks sufficient experimental evidence. Despite the extensive argumentation, the referee remains unconvinced due to a lack of solid scientific backing; the studies and conclusions are overly simplistic for the current stage of research on hybrid organic-inorganic perovskites. At this point in the field's development, the findings will not significantly benefit the scientific community. As a result, the referee does not support acceptance.

Response: We cannot agree with these statements. In our last response letter, we actually provided in-plane device characterizations that demonstrated superior photoelectric performance, highlighting the great application potential of ultrathin hybrid ferroelectrics. Therefore, the statement of "lacks sufficient experimental evidence" is not reasonable and was made without offering any reason. As we explained in our last response letter, it is not necessary to offer detailed device characterization based on a vertical terminal system in a study focused on in-plane ferroelectric materials. The application value of ferroelectric materials is never determined solely by their performance within a specific device structure, such as a vertical system. The comment that our work is "out of touch with the field" is scientifically incorrect at the first place. The absence of detailed vertical device characterization is common among in-plane ferroelectric studies, as well as in the ultrathin ferroelectric studies you mentioned [Science, 379(6638), 1218-1224; Science Advances, 4(7), eaar7720].

The statement that "the findings will not significantly benefit the scientific community" was made without offering any reason. The reviewer has not provided any feedback or engaged in any scientific discussion regarding our explanation of the research significance in our last response letter. As we explained in detail, for ferroelectric semiconductors at the nanoscale, our work helps to address the following issues: 1) low photoelectric signals in ultrathin systems [Science Advances, 4(7), eaat3438]; 2) impractical industrial production due to high synthesis and fabrication costs [Physical Chemistry Chemical Physics, 23(38), 21376-21384; Nature Materials, 22(5), 542-552]; and 3) the scarcity of available 2D ferroelectrics [Nature Materials, 22(5), 542-552]. None of the previous works demonstrate the combined features in other ultrathin ferroelectric semiconductors that can address these issues. Before the submission of this manuscript, research on hybrid perovskite ferroelectrics (HPFs) semiconductors focused on bulk crystals only. The ferroelectric and photoelectric robustness of this promising 2D system have remained in the dark for over a decade. Our work addresses the critical issues related to HPFs as nanoscale ferroelectrics semiconductor from both experimental (demonstrating retained ferroelectricity and photoconductive properties) and theoretical perspectives (limitations of the classic Hecht equation). These issues have never been successfully tackled in any previous studies on HPFs or, in general, on semiconductor ferroelectrics. The significance of our work is thus clear and undeniable.

Merely demonstrating the existence of ferroelectricity at the single-unit cell level for hybrid organic-inorganic perovskites is inadequate. It is crucial to show that monolayer structures perform better in terms of switching or polarization compared to thicker films. Given the semiconductor/insulating nature of these systems, it is expected that ferroelectricity can be maintained, especially when insulative organic cations are present, as the depolarization effect is notably reduced. Numerous studies have already showcased ferroelectricity at the monolayer level in 2D materials.

Switching metrics and polarization can be effectively compared within the same measurement system by varying thickness. Cross-system comparisons can also be made using the same frequency or thickness.

Ultimately, the referee defers to the editor for a final decision based on the provided comments.

Response: We do not agree with these statements and believe that these measurements are unnecessary for the following reasons:

1. We cannot say it is common practice to assess a work not by its impact, but rather by whether the experimental method is simple or complex, or by judging whether the amount of measurement is sufficient regarding a characterization that is not directly related to addressing the main issues of the topic. It is not common practice either to assert certain measurement is crucial without providing any specific reason. Whether there is an improvements or disadvantages (not impractical for electric operation) in switching metrics is not related to the issues we aim to address in this work. If the demonstration of ferroelectricity at the monolayer level is deemed unimportant, then more detailed characterization focused on this aspect shall not be considered as crucial. If fulfilling these measurements would influence the decision-making regarding publication, then the criteria for assessing a work would be based not on research significance but rather on the quantity of measurements performed regarding an unimportant property. We cannot say it is reasonable.

2. We would like to point out these statements contain various scientific mistakes that affect the necessity of these measurements. Firstly, polarization and switching metrics do not necessarily increase with decreasing dimensions. Coercivity can either increase [Nature, 391(6670), 874-877], decrease [ACS Nano, 12(5), 4736-4743], or remain relatively independent [Applied Physics Letters, 89(23)] with dimension scaling. Thus, proposing measurements based on the expectation of improvements in polarization or switching metrics with dimensional scaling is scientifically misguided. If the failure to demonstrate an improvement in switching metrics can be a appropriate reason to dismiss the research, then all aforementioned nanoscale ferroelectric studies without improvements in switching metrics would also need to be rejected, which is clearly not the case. Secondly, it is incorrect to undermine the significance of demonstrating ferroelectricity in HPFs solely because of the presence of an organic component. It is a basic knowledge that the critical thickness—the point at which materials exhibit significant deterioration in ferroelectricity—of

many insulating organic ferroelectrics, such as the benchmark PVDF-TrFE, is substantially larger than that of traditional ferroelectric oxides [Journal of Applied Physics, 89(5), 2613-2616]. Electrical properties are never the only factors influencing the deterioration of ferroelectricity, and being electrically insulating never guarantees the survival of ferroelectricity at the nanoscale. Therefore, it is incorrect to disregard the importance of new discoveries in organic and hybrid ultrathin ferroelectrics because of their organic composition. Additionally, it is well-known that the deterioration of ferroelectricity at the nanoscale is closely related to the effects of interface strain, which may significantly alter the ferroelectric lattice [Nature, 422(6931), 506-509]. Since the Young's modulus of hybrids and organics is orders of magnitude lower, while interface strain can be much greater [Coordination Chemistry Reviews, 391, 15-29], the discovery of ferroelectricity in organic and hybrid ferroelectrics in ultrathin forms should, in fact, hold even greater significance. This is also one of the reasons why the critical thickness of PVDF-TrFE is much larger than that of inorganic materials [Journal of Applied Physics, 89(5), 2613-2616].

3. Most importantly, these measurements are irrelevant to the issues we strive to address for nanoscale ferroelectric semiconductors and are therefore unimportant to this work. As mentioned in our response, the main uniqueness of HPFs nanoflakes lies in their exceptional photoelectric robustness and the survived ferroelectricity that is independent of flake thickness at the nanoscale. If the switching voltage of the nanoflake (5V for monolayer, according to our PFM study) shows no indication of being impractically high compared to other ultrathin ferroelectrics (2V for bismuth oxide [Science, 379(6638), 1218-1224] and 6-7V for In₂Se₃ [Nano Letters, 18(2), 1253-1258], the materials the reviewer mentioned), it is unreasonable to dismiss our work just because detailed switching metrics or polarization characterizations have not been provided, especially when such measurements has never been regarded as necessary in other ultrathin ferroelectric demonstration studies.